# Atmospheric CO₂ inversion reveals the Amazon as a minor carbon source caused by fire emissions, with forest uptake offsetting about half of these emissions

Luana S. Basso[1,2,3], Chris Wilson[4,5], Martyn P. Chipperfield[4,5], Graciela Tejada[2], Henrique L.G. Cassol[2,6], Egídio Arai[2], Mathew Williams[6], T. Luke Smallman[6], Wouter Peters[7,8], Stijn Naus[9,10], John B. Miller[11], Manuel Gloor[1]

[1]School of Geography, University of Leeds, Leeds, LS2 9JT, UK

[2]General Coordination of Earth Science (CGCT), National Institute for Space Research (INPE), São José dos Campos, Brazil

[3]Department of Biogeochemical Signals, Max Planck Institute for Biogeochemistry, Jena, Germany

[4]School of Earth and Environment, University of Leeds, Leeds, LS2 9JT, UK

[5]National Centre for Earth Observation, University of Leeds, Leeds, LS2 9JT, UK

[6]School of GeoSciences & National Centre for Earth Observation, University of Edinburgh, Edinburgh, EH9 3FF, UK

[7]Wageningen University, Environmental Sciences Group, 6708PB, Wageningen, The Netherlands

[8]University of Groningen, Centre for Isotope Research, Groningen, The Netherlands

[9]Meteorology and Air Quality, Wageningen University and Research, The Netherlands

[10]SRON Netherlands Institute for Space Research, Utrecht, The Netherlands

[11]NOAA - Global Monitoring Laboratory, Boulder, Colorado 80305, USA

*Correspondence to*: Luana S. Basso (lbasso@bgc-jena.mpg.de / luanabasso@gmail.com)

**Abstract.** Tropical forests such as the Amazonian rainforests play an important role for climate, are large carbon stores and are a treasure of biodiversity. Amazonian forests are being exposed to large scale deforestation and degradation for many decades. Deforestation declined between 2005 and 2012 but more recently has again increased with similar rates as in 2007/2008. The resulting forest fragments are exposed to substantially elevated temperatures in an already warming world. These temperature and land cover changes are expected to affect the forests and an important diagnostic of their health and sensitivity to climate variation is their carbon balance. In a recent study based on CO₂ atmospheric vertical profile observations between 2010 and 2018, and an air column budgeting technique used to estimate fluxes, we reported the Amazon region as a carbon source to the atmosphere, mainly due to fire emissions. Instead of an air column budgeting technique, we use here an inverse of the global atmospheric transport model, TOMCAT, to assimilate CO₂ observations from Amazon vertical profiles and global flask measurements. We thus estimate inter- and intra-annual variability in the carbon fluxes, trends over time and controls for the period 2010-2018. This is the longest period covered by a Bayesian inversion of these atmospheric CO₂ profile observations to date. Our analyses indicate that the Amazon is a small net source of carbon to the atmosphere (mean 2010-2018 = $0.13 \pm 0.17$ PgC y⁻¹, where 0.17 is the 1-σ uncertainty), with the majority of the emissions coming from the eastern region (77% of total Amazon emission). Fire is the primary driver of the Amazonian source ($0.26 \pm 0.13$ PgC y⁻¹), while forest

carbon uptake removes around half of the fire emissions to the atmosphere ($-0.13 \pm 0.20$ PgC y$^{-1}$). The largest net carbon sink was observed in the western-central Amazon region (72% of the fire emissions). We find larger carbon emissions during the extreme drought years (such as 2010, 2015 and 2016), correlated with increases in temperature, cumulative water deficit and burned area. Despite the increase in total carbon emissions during drought years, we do not observe a significant trend over time in our carbon total, fire and net biome exchange estimates between 2010 and 2018. Our analysis thus cannot provide clear evidence for a weakening of the carbon uptake by Amazonian tropical forests.

## 1 Introduction

The uptake of carbon dioxide ($CO_2$) by plants helps to mitigate global climate change. The land carbon sink is estimated to have offset approximately 25% of all fossil-fuel emissions since 1960 (Friedlingstein et al., 2020). Tropical forests, like those in Amazonia are the largest in the world and have historically been a major component of this land carbon sink. Measurements of aboveground biomass changes indicate an increase in Amazonian old growth forest biomass over time, summing to a total sink of 0.38 (0.28-0.49 95% C.I.) PgC y$^{-1}$ in the 2000s (Brienen et al., 2015). However, the Amazon carbon cycle is affected by both direct (deforestation and degradation) and indirect (climate change) anthropogenic forest disturbances, an example of the latter being a reduction in the forest carbon uptake capacity during drought years (Phillips et al., 2009; Gatti et al., 2014; van der Laan-Luijkx et al., 2015; Alden et al., 2016). A decline in the Amazon carbon accumulation has been observed over 1983 to mid-2011, as a consequence of an increase in tree mortality throughout this period, possibly as a result of greater climate variability and feedbacks of faster growth on mortality, resulting in shortened tree longevity (Brienen et al., 2015).

Human-induced land use and cover change, and forest degradation (driven by fires caused by anthropogenic activity in association with drier conditions and logging) are the main direct disturbances in the Amazon forest (Fawcett et al., 2022; Lapola et al., 2023). These disturbances have been estimated to result in above ground biomass losses of 1.3 ($\pm0.4$) PgC (between 2012 and 2019; Fawcett et al., 2022). Kruid et al. (2021) attributed 56% of the carbon loss in this region during the period from 2003 to 2019 to deforestation, with the remainder (44%) to forest degradation/disturbance (including fire, natural disturbances, drought-induced tree mortality, edge effects, selective logging, and other extractive activities). Over the past 40 years the Amazon forest area has decreased by 17% (MapBiomas, 2020), and degradation (between 1995 and 2017) accounts for around 17% of total forest area (Lapola et al., 2023). Forest fires are associated with a combination of human activities providing the ignition source, and climatic factors which create drier and hotter conditions (Ray et al., 2005). Tropical forests like those in Amazonia are rarely susceptible to natural fires. In general, the forest fires observed in this region result from the leakage of fires from deforested areas to adjacent forests (Aragão et al., 2018). In addition, deforestation and selective logging promotes degradation of adjacent forests, increasing their vulnerability to fires, which could result in further degradation (Aragão et al., 2018). Silva et al. (2020) found that forest fires affect the Amazon forest carbon cycle for at least 30 years after the fires, with just 35% of this emission being compensated by cumulative $CO_2$ uptake of burned forests during this period.

As climate change continues, extreme climate events across the Amazon region have become increasingly common (Gloor et al., 2013). Recently a warming trend in Amazonian annual mean temperature over the last 40 years was reported, where the eastern and mainly southeastern regions showed stronger trends than the global mean trend (Gatti et al., 2021). The largest increases in Amazon temperature were observed for the dry-season months, in addition to a decrease in precipitation of 17% during these months, strongly enhancing the contrast between the dry and wet seasons (Gatti et al., 2021; Haghtalab et al., 2020). The Amazon is estimated to have suffered a substantial carbon loss due to fires caused by the 2015/2016 El Niño drought and heat wave in eastern Amazon; long-term forest plot monitoring reveals that carbon losses remained elevated for up to 3 years (Berenguer et al., 2021). These impacts could have been amplified by human disturbance, which means that human-modified forests may be more susceptible and sensitive to fires (Berenguer et al., 2021).

Recently Gatti et al. (2021) reported new top-down estimates of the Amazon carbon balance covering the period 2010-2018. The Amazonian carbon balance is of interest for two reasons: first to understand how tropical forest productivity and losses fit in the global carbon balance, specifically the substantial global land sink, and second as an indicator of Amazonian forest performance changes over time. Gatti et al. (2021) found a net carbon release to the atmosphere of $0.29 \pm 0.40$ PgC $y^{-1}$, including $0.41 \pm 0.05$ PgC $y^{-1}$ of fire emissions. The net biome exchange (NBE, representing the balance between photosynthesis, respiration, decomposition and excluding fire) compensated for 31% of fire emissions from the atmosphere, yielding a small NBE sink for Amazonia of $-0.12 \pm 0.40$ PgC $y^{-1}$ (Gatti et al., 2021). In addition, Gatti et al. (2021) reported an east–west difference in total flux mainly related to fire emissions, but also highlight that the southeastern Amazon region acts as a net carbon source (total carbon flux minus fire emissions) to the atmosphere. The authors suggest that the historical land use change and the strong climate trends (the temperature increase and decrease in precipitation mainly during the dry season) observed in this southeast region may explain the positive NBE (i.e. a source of C to the atmosphere) in the southeast, as its estimated positive trend, suggesting that increasing temperatures and decreasing soil water availability have a significant impact on the vegetation carbon balance, at least in southeast Amazonia (Gatti et al., 2021).

These estimates were based on nine years of lower-troposphere vertical $CO_2$ and CO profile observations and an air column mass balance technique to estimate fluxes. In essence, the fluxes are estimated as the difference between the air column $CO_2$ content at the site and the location where air enters the Amazon on its path to the site divided by the air travel time from the coast to the site (Gatti et al., 2021; Miller et al., 2007). Estimates based on this approach have uncertainties. For example, we assume well mixed conditions during the sampling. As reported by the authors, the approach does not account for convective process that may result in losses of surface flux $CO_2$ at the top of the profiles (typically 4.5 km a.s.l.) (Gatti et al., 2021). There are also uncertainties in the estimates of background air concentrations (as assumed that remote Atlantic marine boundary layer concentrations represent the partial column entering the coast; Domingues et al., 2020 and Gatti et al., 2021), and we also do not account for diurnal cycles in NBE that may impact the partial column mean $CO_2$.

In order to extract Amazonian surface flux information from the vertical profiles using an independent approach, we apply here a global three-dimensional (3-D) Eulerian offline chemical transport model, TOMCAT (Chipperfield, 2006) and its inverse model, INVICAT (Wilson et al., 2021) to atmospheric $CO_2$ data. We estimate Amazonian surface fluxes between 2010

and 2018 using the $CO_2$ observations from global surface monitoring sites (Lan et al., 2022) and lower-troposphere vertical profiles in Amazonia (Gatti et al., 2021). As this 3-D transport model is global and simulates convective cloud transport processes, some of the uncertainties are reduced compared to the air column budgeting method. To the best of our knowledge the complete 2010-2018 Amazonian vertical profile dataset has not yet been used in 3-D atmospheric transport inversions. INVICAT uses a variational scheme, based on 4D-Var methods used in Numerical Weather Prediction (NWP) (e.g. Dimet and Talagrand, 1986), to minimize the difference between predicted and observed dry air mole fractions. Using this methodology, we quantify fluxes and analyze their seasonal patterns, inter-annual variability and trends for Amazon. We also estimate carbon emissions from fires to constrain the Amazon carbon budget using flux estimates from an independent global inverse modeling based on atmospheric carbon monoxide (CO) measured from space, and relate the carbon fluxes (total, fire and NBE) to climate controls. In Section 2 we describe the inverse modelling approach and describe the observations used, in Sections 3 and 4 we discuss our results, analyze the drivers of $CO_2$ fluxes (as cumulative water deficit, temperature, solar radiation and burned area), cross-validate our model mole fractions with independent Amazon observations and compare our estimates with other previous published Amazonian estimates, mainly with estimates using an air column mass balance technique. Finally, we summarize on the extent to which our results are in agreement with previous Amazon carbon flux estimates.

## 2 Methods

### 2.1 Observations

We assimilate in-situ surface flask observations from global surface observation sites and Amazonian lower-troposphere vertical profiles of $CO_2$ into the TOMCAT inverse atmospheric transport model, for a nine-year period between 2010 and 2018.

### 2.1.1 Amazonian aircraft profiles

We assimilated $CO_2$ observations from 590 lower-troposphere vertical profiles over five sites in Brazilian Amazon (SAN, 55.0° W, 2.9° S; TAB, 69.7° W, 6.0° S; ALF, 56.7° W, 8.9° S; RBA, 67.9° W, 9.3° S; TEF, 66.5° W 3.6° S; Figure 1). Air samples were collected approximately twice per month aboard light aircraft from 4.4 to 0.3 km a.s.l. using automatic samplers between 2010 and 2018 (see Gatti et al., 2021 for more details). All samples were collected between 12:00 and 13:00 local time, when the boundary layer is fully developed and most likely to be well mixed. Samples were measured for $CO_2$ and CO mole fraction with high accuracy and precision at the Greenhouse gas Laboratory at National Institute of Space Research (LaGEE/INPE), Brazil (Gatti et al., 2021, 2014). For the inversions we used the mean concentration of each vertical profile in the planetary boundary layer (PBL) level (below 1.5km a.s.l., levels with higher influence of the surface flux in the concentrations), and the vertical profile free troposphere mean (above 3.5km a.s.l., levels with lower influence of the surface flux in the concentrations, representing better the background concentrations). The vertical profile data used here are available at PANGAEA Data Archiving, at https://doi.org/10.1594/PANGAEA.926834 (Gatti et al., 2021b).

Recently NOAA/GML have found that the $CO_2$ concentration is artificially reduced when air samples with high (> 1.7%) water vapor are pressurized in PFP flasks to 2.7 bar, as a result of condensation (Baier et al., 2020). The LaGEE system has some differences from the NOAA system and, as reported by Gatti et al. (2022), a preliminary study using vertical profiles near Manaus (Amazonas state, Brazil) compared PFP samples measured for $CO_2$ at INPE/LAGEE to onboard measurements from a trace gas flight analyser largely immune to water effects (Picarro model G2401-m) and found depletions in PFP $CO_2$ similar to those from the Baier et al. (2020) study. They also report that this influence is likely greatest near the surface, as humidity increases towards lower altitudes, which means that true $CO_2$ in the lower half of the profiles may be higher than measured (Gatti et al., 2022), meaning that our current fluxes to the atmosphere presented here could be underestimated.

### 2.1.2 Surface flask observations

To estimate carbon fluxes, we also assimilated $CO_2$ global long-term surface data provided by the National Oceanic and Atmospheric Administration's / Global Monitoring Laboratory (NOAA/GML) (Lan et al., 2022) into the inverse model. A total of 72 monitoring site's data (available at <ftp://aftp.cmdl.noaa.gov/data/trace_gases/>) were used, where air samples in flasks are collected weekly to biweekly (Figure 1, Table A1). These measurements have high accuracy (~0.2ppm) and most of the sites are located in the Northern Hemisphere. There are few monitoring sites in tropical regions, which increases the uncertainties of regional estimates in the tropics. Here we reduce these uncertainties for Amazonia with the inclusion of the lower-troposphere vertical profile data.

## 2.2 Model setup

### 2.2.1 Inverse model setup

To estimate the net carbon flux between Amazon and the atmosphere we use the inverse of the atmospheric transport model TOMCAT (Chipperfield, 2006). TOMCAT is a global 3-D Eulerian offline atmospheric chemistry and air constituent transport model, which has been previously used to estimate greenhouse gas emissions (e.g. Wilson et al., 2016, 2021 and Gloor et al., 2018). The INVICAT inversion framework (Wilson et al., 2014) used is based on the TOMCAT model and its adjoint. A detailed description of the TOMCAT model and the 4D-Var inverse method employed by INVICAT are presented in Chipperfield (2006) and Wilson et al. (2014), respectively. A previous study with simulations of sulfur hexafluoride ($SF_6$) and other species comparing different transport models investigated some of the large-scale transport characteristics (Patra et al., 2011), and shows that TOMCAT in general performed well, slightly overestimating the $SF_6$ inter-hemispheric gradient compared to observations, but within the bounds of other transport models.

The forward and adjoint model simulations were carried out globally at 5.6° x 5.6° horizontal resolution, with 60 vertical levels up to 0.1 hPa. Although we did not investigate the uncertainties of the coarse resolution in our estimates, previous $CH_4$ inversion estimates using TOMCAT model with inversions at 2.8° and 5.6° resolution and assimilating GOSAT data showed that the results were robust at both resolutions (Wilson et al., 2021). The inversions were carried out for each year separately

and each completed 50 minimization iterations. In order to better constrain fluxes during the final months of each year, the inversion for each year was actually run for 16 months, from December of the previous year to the end of March for the following year, with the first one and the final three months being discarded from the results, and each inversion was initialized using 3-D fields provided from the correct date in the previous year. The model meteorology (including winds, temperature and pressure data) was taken from the European Centre for Medium-Range Weather Forecasts (ECMWF) ERA-Interim reanalysis (Dee et al., 2011).

The initial conditions for the inversions come from prior $CO_2$ simulations and inversions which began in 1995. For those simulations the initial conditions were included within the state vector and optimised in order to produce an initial global 3D field consistent with observations. For the assimilated observation data from both surface monitoring sites and the vertical profile sites, the model output was linearly interpolated to the correct longitude, latitude and altitude at the nearest model time step. In addition, uncorrelated random errors of 1 ppm were attributed to each observation. In addition, representation uncertainty for each observation was calculated online during the model simulation as the mean difference across the six model grid cells adjacent (2 in z, 2 in x, and 2 in y) to that containing the observation location. The random and representation errors were then combined in quadrature to provide the overall observation uncertainty.

In addition to atmospheric $CO_2$ mole fractions, a priori monthly mean flux values for each grid cell along with a diagonal error covariance matrix for these values were used as input for the inversion calculation. A priori grid cell uncertainties were assumed to be uncorrelated. The result of the inversion is an a posteriori estimate of monthly mean grid cell fluxes and an error covariance matrix. Using TOMCAT, we ran forward a priori and a posteriori flux estimates to simulate atmospheric $CO_2$ air mole fractions. Here we will refer to the mean a priori and a posteriori fluxes and mole fractions as "prior fluxes", "posterior fluxes", "prior mole fractions" and "posterior mole fractions", respectively. In our $CO_2$ inversion estimate fossil fuel flux was fixed and land-biosphere, ocean and fire emissions were optimized. Prior emissions are given grid cell uncertainties of 308% of the prior flux value to give a total global uncertainty based on the Global Carbon Project (Friedlingstein et al., 2020) of 1.7 PgC y$^{-1}$, with a different uncertainty value attributed to land and ocean grid cells. The differentiation was based on assuming the Global Carbon Project (Friedlingstein et al., 2020) total uncertainty estimates of 1.1 and 0.6 PgC y$^{-1}$ for land and ocean global flux uncertainties, respectively.

To derive the uncertainties for the posterior emissions, we followed the approach described by Wilson et al. (2021), where estimates for each year's posterior emission covariance error matrix using cost function gradient values were produced from the limited-memory Broyden–Fletcher–Goldfarb–Shanno algorithm (L-BFGS). We use this to minimize the cost function (Nocedal, 1980), based on the method suggested by Bousserez et al. (2015). Considering that this iterative method estimates the inverse of the Hessian (the second derivative) of the cost function and the off-diagonal elements of the posterior covariance matrix are not included, the posterior errors included here are estimates, with their own reminimg uncertainties (Bousserez et al., 2015).

### 2.2.2 Prior flux estimates

Prior flux estimates include three components and were taken from available bottom-up models and inventories. Fossil fuel emissions are taken from the CDIAC inventory (Boden et al., 1999) and vary each year up to 2016, after which they were scaled to Global Carbon Budget values obtained from Friedlingstein et al. (2020). For estimates of air–sea fluxes we used a combination of Takahashi et al. (2009) and Khatiwala et al. (2009), following the methodology described by Gloor et al. (2018), and they were scaled to the Global Carbon Budget values (Friedlingstein et al., 2020). For the monthly land-biosphere fluxes (net land gains or losses) we used an annually repeating and balanced land vegetation–atmosphere $CO_2$ flux from the CASA GFED4 (Carnegie–Ames–Stanford) land biosphere model (Potter et al., 1993; Randerson et al., 2018), an average climatology for 2003–2013. We did not change the land-biosphere prior annually because we preferred the inter-annual variations to be informed by the atmospheric observations. In CASA model, primary productivity is predicted using the relationship between greenness reflectance properties, the fraction of absorption of photosynthetically active radiation (fPAR) and a light utilization efficiency term, where the canopy greenness is measured using a Normalized Difference Vegetation Index (NDVI) that is computed from the ratio of visible and near-infrared radiation reflected from the canopy as detected by the AVHRR satellite sensor (Potter, 1999).

To evaluate the influence of the Amazon vertical profile data on flux estimates, we have also performed an inversion without the profile data, using only the NOAA surface data. The latter approach was shown previously to induce large biases in the estimated Amazonian fluxes, resulting from a lack of tropical constraints (van der Laan-Luijkx et al., 2015) and an overestimated tropical-NH dipole (Stephens et al., 2007). For simplicity, here we will call the posterior fluxes from the inversion using the Amazon vertical profile data and the inversions without that data as "posterior total flux (with Amazon observations)" and "posterior total flux (without Amazon observations)", respectively.

To evaluate the influence of the biosphere prior on flux estimates, we compare our inversions using the CASA model as land-biosphere prior flux with inversions using the CARbon DAta MOdel FraMework (CARDAMOM) (Bloom et al., 2016) as land-biosphere prior flux for the South America with $1° \times 1°$ spatial and monthly temporal resolutions between 2001 and 2017 (inclusive). CARDAMOM is a Bayesian calibration system that generates diagnostic estimates of the terrestrial C cycle (pools and fluxes) and relevant process parameters. CARDAMOM explores a parameter hyper-volume for a fast-running intermediate complexity model, DALEC, and accepts parameter sets that generate model outputs consistent with observations and their uncertainty. Data used as inputs include time series information on leaf area index (LAI) magnitude and uncertainty, that is extracted from the $1 km \times 1 km$ 8 d product from Copernicus Service Information (2020). Fire and forest biomass removal was imposed using earth observation information. The MODIS burned fraction product (Giglio et al., 2018) determines the areas where fire is imposed. Emissions are determined assuming as the product of the MODIS burned fraction input, the simulated biomass pools (labile, foliage, roots, wood, litter and soil) and tissue specific combustion completeness (CC) parameters. The CC parameters are estimated on a per-pixel basis as part of the CARDAMOM process. As part DALEC's fire model, a fraction of the burned but not combusted biomass pools undergoes mortality (tissue resilience) resulting in the

generation of litter. For details see Exbrayat et al. (2018). Forest biomass removal is imposed using the Global Forest Watch (GFW) forest cover loss product (Hansen et al., 2013). Meteorological drivers are drawn from the Climatic Research Unit and Japanese reanalysis (CRU-JRA) v1.1 dataset, a 6- hourly $0.5° \times 0.5°$ reanalysis (University of East Anglia Climatic Research Unit and Harris, 2019). For more details see Smallman et al. (2021).

### 2.2.3 Estimation of carbon emissions from fires

To estimate the contribution of biomass burning emissions to total carbon emissions from Amazonia, we estimated fire emissions with an independent inversion with TOMCAT/INVICAT by assimilating total column carbon monoxide (CO) values from MOPITT radiometer data (V8) on the TERRA satellite (Deeter et al., 2019) globally. Note, that in this inversion no vertical profile data for the Amazon region were assimilated. Recent studies by Zheng et al. (2019) and Naus et al. (2022) have shown that this approach for deriving fire emissions is complementary to surface remote-sensing based methods. Due to

the high density of available observational data, we carried out this inversion at $2.8° \times 2.8°$ horizontal resolution with 60 vertical levels up to 0.1 hPa. We used uncorrelated prior grid cell emission uncertainties of 450% to give a global annual uncertainty of 15%. The model was sampled at the longitude and latitude of each MOPITT retrieval, and the corresponding averaging kernels were applied to produce a model total column comparable to that of the satellite. For use in the inversion, we took an error-weighted average hourly mean of all retrievals within each grid cell, and applied to these uncorrelated

observation uncertainties of 20% of the observed total column value added in quadrature to the supplied uncertainties. Averaging the observations within each grid cell reduces the need to apply observational error correlations. As prior fluxes we use fire emissions from GFED V4.1s (van der Werf et al., 2017), anthropogenic and oceanic emissions from CMIP6 (Hoesly et al., 2018) and direct biogenic emissions from CCMI (Morgenstern et al., 2017), as the secondary formation from isoprene, assumed to be instantaneous so applied as a surface flux. For secondary formation from methane, monthly mean methane

concentrations were taken from a previous TOMCAT-based methane inversion where the reaction with OH lead directly to CO (Wilson et al., 2021).

    To estimate CO flux from fire, we remove the non-fire CO fluxes from the total CO flux we estimated, by multiplying the CO flux by the prior fire fraction of the total flux in that grid cell. Which means that it is not possible to produce posterior fire emissions in cells which contain no prior fire emissions. Finally, we convert the CO fluxes to carbon fluxes by multiplying the

CO fluxes with a biomass burning emission ratio of 16 (($ppm\ CO_2$)/($ppm\ CO$)), based on the mean $CO:CO_2$ ratio of four Amazon sites estimated by vertical profile measurements by Gatti et al. (2021). Note that these fire $CO_2$ emissions were not used as a fixed prior in the $CO_2$ inversion: instead we subtracted these from the terrestrial non-fossil $CO_2$ flux estimated in the inversion to derive Net Biome Exchange (NBE) of the biosphere.

    To evaluate our carbon fire emission estimate, we compare our $CO_2$ fire flux and NBE flux from our CO TOMCAT-based

inversion with $CO_2$ fire flux estimates based on CO inversion estimates from Naus et al. (2022). For the comparison we used their posterior Amazon biomass burning inversion estimates based on CAMS Global fire assimilation system (GFAS v1.2, Kaiser et al., 2012) as a prior, with the optimized CO emissions assimilating MOPPIT data for the South America domain (for

detailed information about the inversions see Naus et al., 2022). The TM5 model used for these inversions used a nested grid over the Amazon region with horizontal resolution $1° \times 1°$, and 25 vertical levels. Fluxes were optimized on a 3-day basis, and

fire emissions were emitted using vertical distributions from a fire emission model. It should be noted that NBE fluxes calculated based on TOMCAT total carbon fluxes and TM5 fire emissions might have large errors due to the many differences between the methodologies and transport schemes in the two models. We estimated NBE fluxes subtracting these $CO_2$ from fires from the total $CO_2$ flux estimated in our inversion. Note that $CO_2$ fire flux estimates based on Naus et al. (2022) inversions were done using $CO:CO_2$ ratios based on GFAS emission factors for each grid cell. Considering that estimates from Naus et

al. (2022) were done between April to December and for a different Amazon area, for comparison we recalculated our $CO_2$ and CO TOMCAT-based inversions to the same area and time period (April-December over the nine years).

### 2.2.4 Cumulative water deficit (CWD)

As an indicator of plant soil water stress we use climatic cumulated water deficit (CWD). CWD is a monthly soil water balance based on two simplifying assumptions: 0.1 m month$^{-1}$ evapotranspiration and that any excess water runs off. Thus


$$CWD_{i,j}(t) = \begin{cases} 0 \ if \ CWD_{i,j}(t-1) + Precip(t) - 0.1 \ (m \ month^{-1}) > 0 \\ CWD_{i,j}(t-1) + Precip(t) - 0.1 \ (m \ month^{-1}) \ else \end{cases} \tag{1}$$

where $t$ is time (month) and $i,j$ are grid cell indices. Furthermore, assuming that soil is fully recharged during the wettest month, CWD is reset to zero at the month of maximum precipitation, calculated separately for each grid cell as a climatic mean. From the monthly CWD maps, 'maximum climatic water deficit' is defined as the maximum over the 11-month period following the

month with maximum precipitation. We use precipitation estimates provided by TRMM (version 7) (Tropical Rainfall mission, Huffman et al., 2001) which has a $0.25° \times 0.25°$ latitude by longitude spatial resolution.

### 2.2.5 Temperature

For temperature analysis we used 2-m air temperatures from ERA-5 that are monthly means of daily means since 1959 (here

used between 2010 and 2018) and with a resolution of $0.25° \times 0.25°$ latitude–longitude, obtained from the ECMWF (https://cds.climate.copernicus.eu/cdsapp#!/dataset/reanalysis-era5-single-levels-monthly-means?tab=overview; Hersbach et al., 2020).

### 2.2.6 Solar radiation

For solar radiation we used the global monthly mean surface shortwave solar radiation downward flux under all-sky conditions,

between 2010 and 2018, obtained from Clouds and the Earth's Radiant Energy System (CERES-EBAF Ed4.1; https://ceres-tool.larc.nasa.gov/ord-tool/jsp/EBAF41Selection.jsp) at 1° resolution (Loeb et al., 2018; Kato et al., 2018).

**2.2.7 Burned area**

Burned area data was obtained from the Moderate Resolution Imaging Spectroradiometer (MODIS) Collection 6 MCD64A1 burned area product (Giglio et al., 2018). This collection provides monthly tiles of burned area with 500 m spatial resolution over the globe, and was resampled to $1° \times 1°$ spatial resolution. The algorithm to estimate burned area uses several parameters from the Terra and Aqua satellite products, including daily active fire (MOD14A1 and Aqua MYD14A1), daily surface reflectance (MOD09GHK and MYD09GHK), and annual land cover (MCD12Q1) (Vermote et al., 2002; Justice et al., 2002; Friedl et al., 2010).

# 3 Results

## 3.1 Spatial distribution and seasonal pattern of Amazon carbon fluxes

To evaluate how well the inversion fitted the assimilated Amazon vertical profile data we compared the prior and posterior mole fractions with the observations (Figure 2) both for the mean observations from Amazon vertical profiles, both below 1.5km and above 3.5km altitude. Estimated posteriori $CO_2$ mole fractions have a similar magnitude and a positive trend as the observations including the global posterior global mean mole fraction which follows the global trend (Figure A1). In our Amazon mole fractions we observed a large improvement after the assimilation of observations in the model: the mean difference between estimated mole fraction and observations was reduced by 57% and 49% for the mean mole fractions below 1.5km and above 3.5km altitude, respectively (Figure 2 and Table A2). In addition to a decrease in residuals, we also found higher correlations between the observations and the posterior mole fraction compared to the difference between observations and prior mole fractions (Figure 2).

In Figure 3 we display the 2010-2018 quarterly and annual mean prior total, posterior total and posterior fire carbon flux distributions in the Amazon region, to show the long-term flux distribution over this period. The nine-year mean prior flux distribution shows a source of carbon to the atmosphere during the first quarter of the year (January-March) in the west-central region, while a sink of carbon between July to December, mainly occurring between July to September i.e. during the dry season. After assimilating the Amazon vertical profile data, the posterior fluxes had a different seasonal pattern, with a significant sink in the central Amazon during January and March and a source to the atmosphere in the western region. In addition, a carbon source to the atmosphere was estimated in the eastern Amazon from July to September, which is consistent with the nine-year mean carbon emissions from fires estimated in this region over this time based on the CO inversions using MOPITT data and with the drought period in Amazon region (Figure 3c and d).

Our data reveal distinct spatial and seasonal carbon flux patterns in the nine-year monthly means and a significant change in posterior fluxes when vertical profile data were assimilated in the model (linear regression between posterior flux with Amazon data and prior flux: r = 0.13 and p = 0.16). Posterior total fluxes obtained without assimilating the Amazon vertical profile data result in a similar seasonal pattern as the prior total flux (linear regression between posterior flux without Amazon data and

prior flux: r = 0.66 and p < 0.05), mainly between January and March, showing the Amazon as a source of carbon to the atmosphere (Figure A2). This is in contrast with the posterior total flux estimates when the Amazon vertical profile data are

assimilated in the inversions. The posterior total flux without the Amazon vertical profile data also shows an uptake of carbon during May and June similar to the prior total fluxes, but with a reduction in the magnitude of these fluxes, particularly in the eastern Amazon (Figures 3 and 4). These results indicate the strong influence and thus importance of Amazonian regional data in the inversions to constrain the Amazon carbon fluxes estimates, as also found by van der Laan-Luijkx et al. (2015) and Botía Bocanegra (2022).

Large carbon emissions from fires were observed in Amazonia from August to December, mainly from the south and east regions (Figures 3, 4 and 5). Fires also contribute to emissions to the atmosphere between January and March, but mainly from the western-central region, due to fires occurring in the Northern Hemisphere (Figures 3, 4 and 5).

To estimate the $CO_2$ net biome exchange (NBE) we subtracted the fire emissions calculated using the estimated CO fluxes by TOMCAT inverse modelling (Figure A3), from our posterior total fluxes (Figure 4 and 5 and Figures A4 and A5). Our NBE

represents the balance between photosynthesis and respiration. We use the following sign convention: positive NBE is a flux to the atmosphere. According to our results the forest, not considering fire emissions, is a sink during the wet season and still acts as a sink in part of the dry season, except in July and October (Figures 3 and 4). This dry season sink compensates part of the carbon emissions from fires, but with the sink located mainly in the western-central Amazon (Figure 3). During the years with strong droughts such as 2010 and 2015-16, a reduction in this dry-season uptake (near neutrality) was estimated (Figure

4, A4 and A5, discussed in detail in Section 4). In the western-central region we estimate a positive NBE flux to the atmosphere between April and June, which could be caused by emissions from decomposition processes (Figure 4 and A4), as the carbon emissions due to dead wood decay in the following years of a burning event (Silva et al., 2020; Anderson et al., 2015). This result resembles the seasonal cycle of NBE found by Botía et al. (2022), who used ATTO-tower $CO_2$ time series data to find NBE rapidly declining at the end of the wet season, resulting in a source of $CO_2$ in June. In the eastern region we also estimate

positive NBE fluxes (during June and July) that could be related to the decomposition process, but these emissions have lower magnitude than the observed in the western-central region. We highlight that the southern border of the Amazon region is characterized by a steady transition from the Amazon lowland rainforest to a mainly non-forested landscape, with predominately open vegetation types, such as savannas (campos cerrados and alluvial flooded savannas), savanna woodlands (cerradão) and other scrubby vegetation types (Eva and Huber, 2005). It also includes deforested areas with land use change

conversion (Figure A6).

We also investigated the possible relation of climate conditions with the intra-annual variability in total $CO_2$ fluxes. An increase in the net carbon loss to the atmosphere was observed during warmer (r= 0.34, and Student's T-test p <0.05) and drier (r= 0.61, p <0.05) periods, during which also solar radiation (r= 0.20, p <0.05) and burned area (r= 0.22, p <0.05) increased. Linear regressions between posterior monthly mean fire fluxes and temperature, CWD, solar radiation and burned area all reveal

significant correlations (r= 0.61, p <0.05; r= 0.33, p <0.05; r= 0.52, p <0.05; and r= 0.86, p <0.05, respectively), (Figures A7 and A8). Furthermore, an increase in total and fire emissions was estimated during the strong drought years (2010 and 2015–

16) as expected. Note that the inter-annual variability in posterior $CO_2$ total fluxes is driven by the Amazon aircraft observations alone, as the land-biosphere prior flux is climatological (i.e. the same for every year) over the period.

No significant relationships between monthly posterior NBE fluxes and climate variables were observed (Figure A9). We also investigated the correlation between NBE fluxes and climate variables with a time lag (one, two and three months of lag) but no significant correlation was observed. For western-central and eastern Amazon regions we found a similar relation between posterior fire fluxes and climate conditions as what was observed for the Amazon as a whole (Figures A4, A5 and to A8).

## 3.2 Amazon carbon balance and its inter-annual variability

When the data from the aircraft vertical profiles were assimilated in the inversions the posterior total flux estimates over the period from 2010 to 2018 (including fire emissions) of $0.13 \pm 0.17$ PgC $y^{-1}$ are positive, with the majority of the emissions coming from the eastern region ($0.10 \pm 0.08$ PgC $y^{-1}$; Table 1). A larger emission to the atmosphere was estimated by the inversions when only NOAA surface site data were assimilated (without the data from the Amazon vertical profiles) resulting in a total emission of $0.48 \pm 0.17$ PgC $y^{-1}$ (including fire emissions). Fire emissions are the main reason for the flux to the atmosphere over the period, $0.26 \pm 0.13$ PgC $y^{-1}$, with the largest contribution also coming from the eastern region (Table 1). Part of these fire emissions are compensated by the forest uptake in both western-central and eastern Amazon regions (72% and 33% of the fire emissions, respectively). We highlight that the Amazon region is a carbon source to the atmosphere when we include fire emissions over this period, with an uptake by the forest (NBE flux) that compensates 50% of the fire emissions. Linear regressions between annual mean posterior total flux and temperature, CWD, solar radiation and burned area yield significant correlations: r= 0.55, p 0.12; r= 0.62, p 0.07; r= 0.54, p 0.13, and r= 0.50, p 0.17, respectively. These annual mean correlations are driven mainly by the drought years, 2010 and 2015-2016. In addition, we found similar relationships between annual mean posterior fire flux and temperature, CWD, solar radiation and burned area (r= 0.75, p <0.05; r= 0.68, p <0.05; r= 0.56, p 0.12, and r= 0.84, p <0.05, respectively), (Figure 5, A10 and A11). However, we did not find any significant relationships between annual mean posterior NBE flux and climate variables (temperature, CWD and solar radiation; Figure A12). Note that our total emission estimates could be over or underestimated during 2015 and 2016, because of the low number of vertical profile data available for this period (Figure A13).

$CO_2$ flux estimates over our nine-year study period indicate that Amazonian total, NBE, and fire emissions do not exhibit significant time trends, neither for the western-central nor eastern regions (Figure 6).

## 3.3 Sensitive tests

We also estimate Amazonian $CO_2$ fluxes using our atmospheric inversion but replacing the biosphere prior flux estimates of CASA by the estimates of CARDAMOM for the South America region (Figure A14). We also observed a large improvement after the assimilation of observations in the model as for the inversions using CASA as prior flux estimates (Figure A15 and Table A2). Comparing both estimates (from CARDAMOM and CASA models) of land-biosphere fluxes used as prior in the

inversions, we found that CARDAMOM shows a large carbon uptake (prior total flux of -2.50 ± 0.43 PgC y$^{-1}$) for the Amazon

region in contrast to the estimates from CASA model (prior total flux of 0.08 ± 0.24 PgC y$^{-1}$). CARDAMOM prior flux estimates show a large carbon sink in Amazon between January and March in contrast with a carbon source to the atmosphere estimated by CASA model. The large uptake was not reproduced after the assimilation of Amazon observational data. After assimilating the Amazon vertical profile data in the inversions using CARDAMOM as the land-biosphere prior, the posterior estimate shows a strong reduction in the uptake for the Amazon region (posterior total flux of -0.19 ± 0.17 PgC y$^{-1}$) compared

to the prior (Figure A14). This result shows that the large land biosphere sink estimated by CARDAMOM is inconsistent with the Amazon atmospheric vertical profile data. Although the inversion using CARDAMOM as a prior estimates the Amazon to be a small overall carbon sink, while the inversion using CASA model as a prior estimates the Amazon to be a small source to the atmosphere (0.13 ± 0.17 PgC y$^{-1}$), the intra-annual seasonality from both inversions is similar (Figure A14). Also, both posterior estimates have a similar spatial flux distribution. Posterior flux estimates using CARDAMOM as land-biosphere

prior flux also estimates the eastern Amazon to be a carbon source to the atmosphere from July to September, and during January and March to be a significant sink in the central Amazon, while the western region to be a source to the atmosphere (Figure A14).

We compared fire fluxes based on CO inversion estimates of Naus et al. (2022) and estimated NBE fluxes subtracting these $CO_2$ from fires from the total $CO_2$ flux estimated in our inversion, with our estimates based on TOMCAT CO inversions. We

found similar intra- and inter-annual variability and flux magnitudes when compared to our NBE and fire estimates based on TOMCAT CO inversions with estimates based on their CO inversions (Figure A16 and Table A3). Both CO inversions assimilated the same MOPITT observations, but differ in inversion methodology, model transport and in the emission factor to convert CO flux from fires to $CO_2$ flux. Some difference in both estimates (for both fire and NBE fluxes) could be related with these differences in both approaches. To get a truly independent estimate of NBE from another inversion model, it would

need to estimate both total carbon and fire carbon. Also, both $CO_2$ fire emissions (based on the two different CO optimized fluxes) could be used as prior in a future $CO_2$ inversion to investigate the dependence of the fires estimates in the NBE optimization.

### 3.4 Comparison to independent observations

To validate our inversions results we used independent in situ observations of $CO_2$ mole fraction in the Amazon region made at the Amazon Tall Tower Observatory (ATTO) site (2.14°S, 58.99°W, measurements at 80m height; Barbosa et al., 2022 and Botia et al., 2022) and with vertical profile data measured at Manaus (MAN) (2.59°S, 60.21°W with profiles extending from approximately 0.2 to 5km height above ground; Miller et al., 2021). For the comparison between modeled and observed mole fractions, the model data was sampled at the grid cell closest to the site locations.

The ATTO timeseries is based on observations made between 2012 and 2018 (Barbosa et al., 2022), and the data presented here are available upon request at <https://attodata.org> (last access: 11 May 2023). We calculate the monthly mean mole

fractions based on only daytime dry air mole fractions (13:00−17:00 local time), which were representative of well-mixed convective conditions (Botia et al., 2022). In addition, for the year 2015 we remove from the comparison the months without vertical profile data assimilated in the inversion. Although the ATTO tower measurements are made 80m above ground, i.e.

quite close to the ground, in general we found good agreement between model and observations (Figure A17), with a reduction of the bias after the inversion from 0.9 (range of -3.9 to 7.7) ppm to 0.3 (range of -5.3 to 4.7) ppm (t-test: p <0.05).

In addition, we compare the model mole fractions to the aircraft vertical profiles in MAN above 3.5km and below 1.5km (Figure A18). The data record from MAN for the same period of our inversions is for the years 2017 and 2018. Flights are undertaken approximately every 2 weeks and in general measurements were taken between 12:00 and 13:00 local time, when

the boundary layer is fully developed. We found a reduction in the bias between model and observations after the inversions, for the mean below 1.5km from -0.3 (range of -6.5 to 6.6) ppm to 0.2 (range of -4.3 to 5.0) ppm (t-test: p = 0.17); and for the mean above 3.5km height from -0.1 (range of -3.1 to 2.1) ppm to -0.4 (range of -1.9 to 0.5) ppm (t-test: p = 0.13). We also found a reduction of the mean bias of the difference between the mean below 1.5km and the vertical profile free troposphere (above 3.5km), from -0.2 (range of -5.4 to 7.6) ppm to 0.6 (range of -4.2 to 5.6) ppm (t-test: p = 0.08). The posterior

comparisons also show an increase in the bias close to surface, which means that the local sources close to this site might be overestimated at this model resolution, there be errors in the model's representation of vertical mixing or a positive bias in the assimilated Amazon vertical profiles in this region may remain. However, in general the cross-validation with observations from ATTO tower and MAN vertical profiles showed an improvement in the model bias and temporal variation after the assimilation of Amazon vertical profile observations.

**4 Discussion**

The posterior fluxes when vertical profile data were assimilated in the inversions led to a change compared to the prior in the fluxes seasonal cycle, and additionally showed a larger reduction in Amazon total emission in comparison with the posterior fluxes when just NOAA surface data were assimilated (Figures 3 and 4 and Table 1). This once again highlights the importance of assimilating regional data in the inversions to better constrain the tropical forest fluxes (van der Laan-Luijkx et al., 2015;

Alden et al., 2016; Botía et al., 2022). This result is not dependent on the assumed prior sources and sinks, as we also found a significant reduction of the large land biosphere carbon uptake suggested by CARDAMOM for the Amazon region after assimilating the Amazon vertical profile data in the inversion (Figure A14).

Using CASA model predictions as land-biosphere prior flux we estimate the Amazon region to be a total (i.e. including emissions from fire) net source of C of $0.13 \pm 0.17$ PgC y$^{-1}$ over our analysis period. The largest emission comes from the

eastern Amazon and from the transition region between the forest and the Cerrado area (as shown in Figure 3), while the largest uptake was observed in the western-central region. Our results indicate that the Amazon is a source of carbon to the atmosphere caused by fire emissions, which were larger than the estimated Amazon land sink, but we highlight that during this period the forest uptake removes around half of the fire emissions to the atmosphere.

Globally, the land $CO_2$ sink was estimated to be $3.1 \pm 0.6$ PgC y$^{-1}$ during the decade 2011–2020 (29 % of total global $CO_2$ emissions, Friedlingstein et al., 2022), and continued to increase during this period likely in response to increased atmospheric $CO_2$ (Friedlingstein et al., 2022). However, the land sink varies strongly inter-annually, with decreased land carbon uptake during El Niño events. According to Friedlingstein et al. (2022), in general the tropical region (30° S–30° N) has a carbon balance close to neutral over the 2011–2020 period, however the tropical region is most strongly correlated with inter-annual variation of atmospheric $CO_2$ (Friedlingstein et al., 2022). Note that this tropical region estimate did not include the information provided by the Amazon vertical $CO_2$ profile data we used here. The Tropics is also where the largest land-use emissions occur, including the Arc of Deforestation in the Amazon basin (Friedlingstein et al., 2022). We did not observe an increasing trend over time in the land carbon uptake for the Amazon region, in contrast to the continued increase in the global land sink reported by Friedlingstein et al. (2022).

Based on a distributed network of 321 forest survey plots from RAINFOR Brienen et al. (2015) estimated a 30% decrease in the total net carbon sink into intact Amazon live biomass from 0.54 PgC y$^{-1}$ (95% confidence interval 0.45–0.63) in the 1990s to 0.38 PgC y$^{-1}$ (0.28–0.49) in the 2000s was estimated. Phillips and Brienen (2017), based also on the RAINFOR network plot measurements, estimated an Amazon-wide forest biomass carbon sink between 1980 and 2010 of 0.43 PgC y$^{-1}$ (CI 0.21-0.67). Finally, Hubau et al. (2020) reported a decrease in the Amazon carbon net sink in the last decades, from 0.68 PgC y$^{-1}$ (CI 0.54–0.83) between 1990 and 2000 to 0.45 PgC y$^{-1}$ (CI 0.31–0.57) between 2000 and 2010, predicting a net carbon sink of 0.25 PgC y$^{-1}$ (CI –0.05–0.54) between 2010–2020. Our posterior NBE estimates (a sink of $0.13 \pm 0.20$ PgC y$^{-1}$) are fairly consistent with the RAINFOR results with regards to magnitude but not with trend over time regarding observed carbon uptake, considering that the areas used for the estimates are different, and that our NBE represents not only the uptake from forest but also non-fire emissions (as decomposition and degradation emissions.

Our posterior fire emissions agree with fire emission estimates for Brazilian Amazonia reported by Aragão et al. (2018), with a total fire emission of $0.21 \pm 0.23$ PgC y$^{-1}$ over the period 2003–2015, based on the relation between MOPITT CO total column and burned forest and deforestation gross $CO_2$ emissions data (Aragão et al., 2018). Recently, Silva et al. (2020) reported that forest fires contribute cumulative gross carbon emissions of ~126 MgCO$_2$ ha$^{-1}$ for 30 years after a fire event, with a mean annual efflux of 4.2 MgCO$_2$ ha$^{-1}$ y$^{-1}$ and emissions from the decomposition of the dead organic matter accounting for ca. 58% (47.4 MgCO$_2$ ha$^{-1}$) of total cumulated net emissions. Van der Werf et al. (2010) estimated that fires were responsible for an annual mean global carbon emission of 2.0 PgC y$^{-1}$ (for the period 1997–2009) with significant inter-annual variability, where about 15% (0.29 PgC y$^{-1}$) was associated with South American emissions mainly from the Southern Hemisphere of South America (14%; 0.27 PgC y$^{-1}$), according to estimates from the Global Fire Emission Data set (GFED V.3). Note that this South American emission estimate was related to a larger area than our Amazon region estimates.

We found clear intra-annual seasonality in our posterior total, fire and NBE fluxes. In general, we found over these nine-years that the Amazon is a carbon sink during November to March (wet season) and also during August and September removing part of the fire emissions during the dry season (Figures 4 and 5 and Figures A4 and A5). Although we did not find a significant relation between our NBE seasonality and the climate variables analyzed (CWD, temperature and solar radiation), our NBE

emission seasonality show good agreement with the Amazon mean net ecosystem exchange (NEE) seasonality measured at five eddy covariance forest tower sites located in the Brazilian Amazon, Manaus forest (K34; 1999–2006), Santarém forest (K67; 2001–2005, 2008-2011 and 2015-2019), forest of Caxiuana (CAX; 1999-2003), Reserva Jarú southern forest (RJA; 2000-2002) and the seasonal inundated forest of Bananal (JAV; 2003-2006) (Gatti et al., 2021c). Our fire emission estimates showed the largest increase during the dry season months of August to October, in agreement with the increase in CWD, temperature, solar radiation and burned area (Figure 5 and Figures A4, A5 and A8).

We found that our total and fire emission estimates inter-annual variability correlates with climatic variations, with larger emissions occurring during hotter and dryer years as in 2010 and 2015-16. This inter-annual variability in our estimates is primarily driven by the atmospheric vertical profile data and MOPPIT CO columns, and not by prior estimates, as in our approach the land flux prior estimates are the same for all years. In 2010 the increase in carbon emissions was mainly caused by an increase in emissions in the western-central region, related to a large increase in fire emissions (2010 flux of $0.32 \pm 0.14$ PgC y$^{-1}$ and a nine-year mean of $0.11 \pm 0.10$ PgC y$^{-1}$; student t-test: $p = 0.14$) and also a reduction of the uptake in relation with the nine-year mean (2010 flux of $-0.04 \pm 0.20$ PgC y$^{-1}$ and a nine-year mean of $-0.08 \pm 0.18$ PgC y$^{-1}$; $p = 0.43$). We also observed an increase in fire emissions in eastern Amazon region during this year, but it is still lower than in the western-central region (2010 flux of $0.28 \pm 0.15$ PgC y$^{-1}$ and a nine-year mean of $0.15 \pm 0.11$ PgC y$^{-1}$; $p = 0.21$). These results are in agreement with the increase in burned areas observed in western-central and eastern Amazon regions for the same period when compared with the nine-year mean (104 and 89% in western-central and eastern Amazon regions, respectively), and with an increase of 7% in the CWD compared with the nine-year mean in the western-central region. Although some p values are larger than 0.05, these results suggest changes in the carbon cycle in 2010. High correlations between soil moisture and MOPITT-derived fire emissions were also reported by Naus et al. (2022) for Amazonas province, confirming the important role of the moisture state of the underlying forest soils.

On the other hand, during 2016 the increase in carbon emissions was mainly related to a reduction in the forest carbon uptake in the Amazon region. Organic land carbon pools (forests, soils) were a net source to the atmosphere during this year (NBE flux of $+0.12 \pm 0.20$ PgC y$^{-1}$; student t-test: $p = 0.14$), while fire emissions increased 61% in the western-central Amazon in relation to the nine-year mean (2016 flux of $0.19 \pm 0.13$ PgC y$^{-1}$ and a nine-year mean of $0.11 \pm 0.10$ PgC y$^{-1}$; student t-test: $p = 0.17$). These indications of reductions in the carbon uptake could be related to hotter and dryer conditions in the western-central region, with an increase of 10% in the CWD in relation to the nine-year mean, and an increase of 0.3 and 0.4 °C in the annual mean temperature in relation with the nine-year mean (the largest positive anomalies in the nine years for both regions) in the western-central and eastern Amazon region. Recently, Fancourt et al. (2022) reported that background climate and soil conditions had a greater influence than the climatic anomalies on Amazon forest photosynthesis spatio-temporal variations, but with the northwestern forests being the most sensitive to precipitation anomalies during the 2015/16 El Niño period.

Gloor et al. (2018) reported a net flux anomaly from the Amazon of $0.5 \pm 0.3$ PgC during the 2015/16 El Niño event (between September 2015 and June 2016), based on previous inversions using TOMCAT and assimilating the Amazon vertical profile data. Our posterior total estimates showed a net flux anomaly for this period of $0.58 \pm 0.20$ PgC for the whole Amazon, with

0.32 ± 0.19 PgC and 0.26 ± 0.09 PgC for the western-central and eastern Amazon, respectively. The majority of the anomalies observed come from a reduction in the carbon sink making NBE fluxes positive in the western-central Amazon with a total net emission of 0.09 ± 0.22 PgC y$^{-1}$ (while the nine-year means for this period show an uptake of 0.04 ± 0.15 PgC y$^{-1}$; p = 0.25), acting as a net carbon source to the atmosphere during this period, in addition to increase in fire emissions at both western-central (flux of 0.23 ± 0.14 PgC y$^{-1}$ for this period while a nine-year mean of 0.11 ± 0.10 PgC y$^{-1}$; p = 0.07) and eastern regions (flux of 0.33 ± 0.14 PgC y$^{-1}$ for this period while a nine-year mean of 0.14 ± 0.10 PgC y$^{-1}$; p = 0.13). Koren et al. (2018) and van Schaik et al. (2018) suggested a reduction in gross primary production, resulting from combined heat- and soil moisture stress, to be a dominant mechanism in agreement with the results in Gloor et al. (2018) based on solar induced fluorescence data measured from space.

While agricultural and deforestation fires are more closely associated with human actions than with climate (Anderson et al., 2018), forest fires are associated with a combination of human activities which provide the ignition source and climatic factors creating dry conditions (Berenguer et al., 2021). During strong drought conditions, such as the drought of 1997/98, fires could escape from agricultural fields and burn standing primary forests that were once considered impenetrable to fire (Brando et al., 2020). A warming trend is being observed in Amazonia, evident since 1980, and it is enhanced since 2000, a period where strong droughts occurred in 2005, 2010, and 2015/16 (the increases in temperature varies with the dataset, time period and spatial scale of the analysis) (Marengo et al., 2021). Also, warming was observed in the eastern Amazon and especially southeastern Amazon, at a rate almost twice as high as the western Amazon (Marengo et al., 2021). Our CWD analysis for Amazonia shows a weak drying trend for almost all regions between 1998 and 2019 (Figure A19). The observed climate tendencies in Amazonia can be different in the western and eastern regions, and the projected changes suggesting a drier and warmer climate in the east, while in the west rainfall is expected to increase in the form of more intense rainfall events (Marengo et al., 2021).

The increase in climate variability impacts both the Amazonian forest (Anderson et al., 2018) and savannah biomes, increasing tree mortality (Aragão et al., 2018) and ecosystem vulnerability to fire (Anderson et al., 2018; Silva Junior et al., 2019). The increased variability, in combination with deforestation, has changed the forest's resilience to fires, in particular in the southern Amazon, where remaining forests have become drier and thus vulnerable to wildfires during recent droughts (Brando et al., 2020). Our posterior fire estimates showed the largest emissions in the eastern Amazon region with an increase in emissions during strong drought years, but we do not find a significant trend over the 2010 to 2018 period. Eastern Amazon is more disturbed by human activity than the western-central region, with larger deforested areas also converted to agriculture and grassy areas (Figure A6).

The clear seasonality in our posterior total, fire and NBE fluxes is generally in agreement with that reported by Gatti et al. (2021), based on a mass balance technique for the Amazon region as a whole, and also for western and eastern regions (Figure A20). It is important to highlight that we use a larger Amazon area in comparison with Gatti et al (2021). Here we included a transition region between Amazon forest and Cerrado biome in the southeast of Amazonia (Figure A6). For the eastern Amazon, the seasonality of the NBE estimate of the two approaches is more similar than the seasonality of the fire emissions.

Gatti et al. (2021) estimated fire emissions to occur during January to March, mainly in the northeastern region, while we did not estimate fire emissions during this period. Part of this difference could be related to the different regions considered as eastern Amazonia in the two studies. The region-of-influence of fluxes influencing $CO_2$ measured at the vertical profiling site (ALF and SAN sites) reported by Gatti et al. (2021), estimated using quarterly mean back-trajectories, has contributions from the North Hemisphere Amazon during this time, an area not considered in our Eastern Amazon region definition. The difference could also be related to the burned areas fraction from the prior flux (from GFED V4.1s) which we multiplied to the CO total flux in each grid cell to derive the CO fire emissions in our inversion, in the absence of burned area fraction will result in no fire emissions in the area, consequently underestimating carbon emissions caused by fire in this region. On the other hand, fire emissions during this period are observed in both approaches in the western-central region. The main difference observed in the estimates for this region was in the NBE during the dry season months of August and September, where our posterior estimates showed an uptake while the mass balance technique estimates (Gatti et al., 2021) showed a source to the atmosphere (Figure A20). A substantial dry season sink in the western Amazon was independently derived from ATTO-tower $CO_2$ observations by Botía et al. (2022), supporting our findings here.

No significant trend over time (between 2010 and 2018) was observed in our posterior emissions, in contrast with the trend in NBE fluxes for the east Amazon region, with an increase in emissions over this time reported by Gatti et al. (2021). Our results indicate that Amazonia is a source of carbon to the atmosphere because of fire emissions, corroborating the findings of Gatti et al. (2021). Our nine-year mean total posterior emissions for the Amazon region are 33% smaller than their total emission estimates, with the largest difference being observed in the eastern region (Figure 7). The largest differences are mainly related with the fire emission estimates, while our posterior NBE estimate represents 90% of their estimates. However, considering the range of both Amazon flux estimates we find similar emissions (Figure 7).

## 5 Conclusions

Our global inverse model estimates of $CO_2$ emissions using Amazon atmospheric vertical profiles and surface observations has allowed us to estimate that over the nine years 2010-2018 the Amazon region acted as a small carbon source to the atmosphere, with a total emission of $0.13 \pm 0.17$ PgC $y^{-1}$. The emissions were greater in eastern Amazonia ($0.10 \pm 0.08$ PgC $y^{-1}$) than in the western region, mostly because of larger fire emissions. The forest uptake (NBE) compensated approximately 50% of the fire emissions and was larger in the western-central region than in the eastern Amazon region (72% and 33% of the fire emissions, respectively). This highlights the importance of public policies to prevent deforestation and fire occurrences to reduce Amazon carbon emissions to the atmosphere and help to mitigate the effects of climate change.

Our estimated carbon fluxes were larger during the extreme drought years such as 2010, 2015 and 2016, mainly because of an increase in fire emissions and a reduction in carbon uptake. However, we did not find any significant trend in carbon emissions over the period 2010-2018. Our analysis thus cannot provide clear evidence for a weakening of the carbon uptake by Amazonian tropical forests.

The inter and intra-annual seasonality of the results from our inversion are in agreement with previous studies (e.g. Gatti et al., 2021; Botía et al. 2022; and Naus et al. 2022). Our study shows the benefit of using regional vertical $CO_2$ profile data over land to better constrain carbon emissions in tropical forests such as the Amazon, thereby improving the estimated magnitude and intra-annual seasonality of the emissions. In turn, this helps to improve global estimates and understand possible climate and human disturbance feedbacks in the carbon cycle.

## 6 Authors contributions

LSB, CW, MG and MPC designed the methodology. LSB wrote the first version of the manuscript and performed the analysis and $CO_2$ inversions. CW performed the TOMCAT CO inversions using MOPPIT data. GT provided the land use change data. HLGC and EA provided the burned area data. MW and TLS provided the CARDAMOM flux estimates. WP and SN provided the CO estimates for the sensitive test. All authors contributed with analysis and text.

## 7 Competing interests

The authors declare that they have no conflict of interest.

## 8 Code and data availability

Posterior CO fluxes from the atmospheric inversion upon request to Chris Wilson. CARDAMOM dataset upon request to Luke Smallman. Prior and posterior mean South American $CO_2$ fluxes on the TOMCAT model grid are available from PANGAEA Data Archiving, the data DOI is https://doi.org/10.1594/PANGAEA.960593.

## 9 Acknowledgements

LSB acknowledge the financial support from São Paulo Research Foundation - FAPESP (2011/17914-0, 2018/14006-4, 2019/23654-2). HLGC acknowledge the financial support from FAPESP (2018/14423-4, 2020/02656-4). GT acknowledge the financial support from FAPESP (2018/18493-7). We acknowledge the support for this work from NERC grants AMAZONICA NE/F005806/1, BIO-RED NE/N012542/1 and ECOFOR NE/K01644X/1. CW and MPC acknowledge contributions from NERC National Centre for Earth Observation (NCEO). HLGC acknowledge the financial support from FAPESP (2020/02656-4). TLS and MW were funded by the UK's National Centre for Earth Observation. The CARDAMOM analyses made use of the resources provided by the Edinburgh Compute and Data Facility (EDCF, http://www.ecdf.ed.ac.uk/). The Amazon vertical profile database was funded by: State of Sao Paulo Science Foundation - FAPESP (2016/02018-2, 2011/51841-0, 2008/58120-3, 2018/14423-4, 2018/18493-7, 2019/21789-8, 2019/23654-2), UK Natural Environmental Research Council (NERC) AMAZONICA project (NE/F005806/1), NASA grants (11-CMS11-0025, NRMJ1000-17-00431, NNX17AK49G), European

Research Council (ERC) under Horizon 2020 (649087), 7FP EU (283080), MCTI/CNPq (2013), CNPq (134878/2009-4). We thank the LaGEE/INPE team (the PI Dr. Luciana Gatti and the colleagues Luciano Marani, Caio C.S.C. Correia, Lucas G. Domingues, Raiane Neves and Stéphane Crispim) who provide the $CO_2$ data from the vertical profiles. In addition, we thank the pilots and technical team at aircraft sites who collected the air samples. We thank numerous people at NOAA/GML who provided the global station network $CO_2$ data, and the MOPITT Team who provides the CO total column data.

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

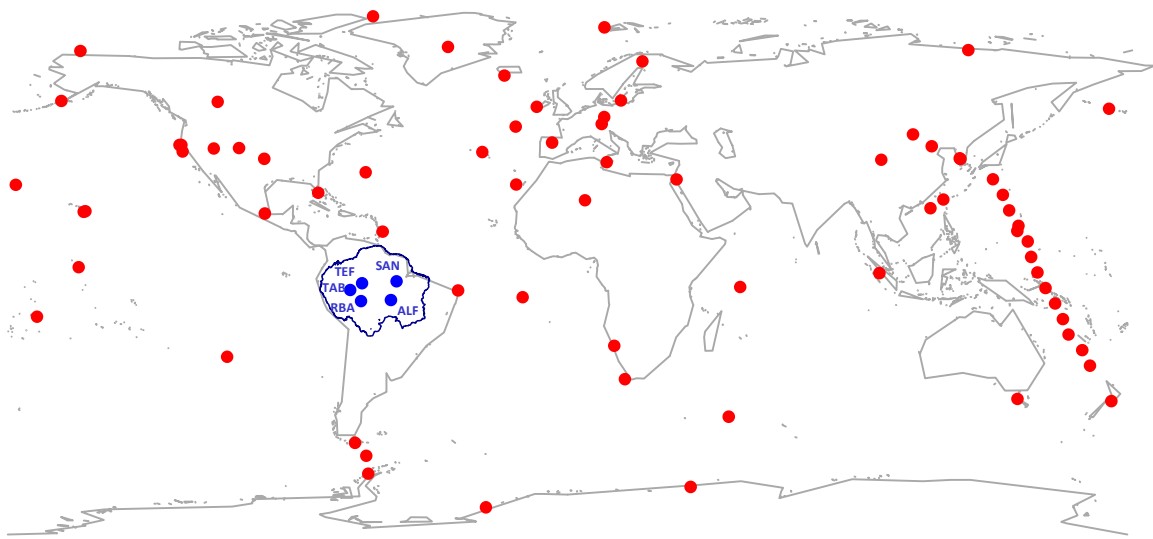

**Figure 1: Locations of INPE/LaGEE Amazon vertical profile sites (blue circles) and NOAA surface sites from which flask-based measurements of CO₂ are assimilated (red circles). The blue contour represents the Amazon area based on Eva and Huber (2005).**

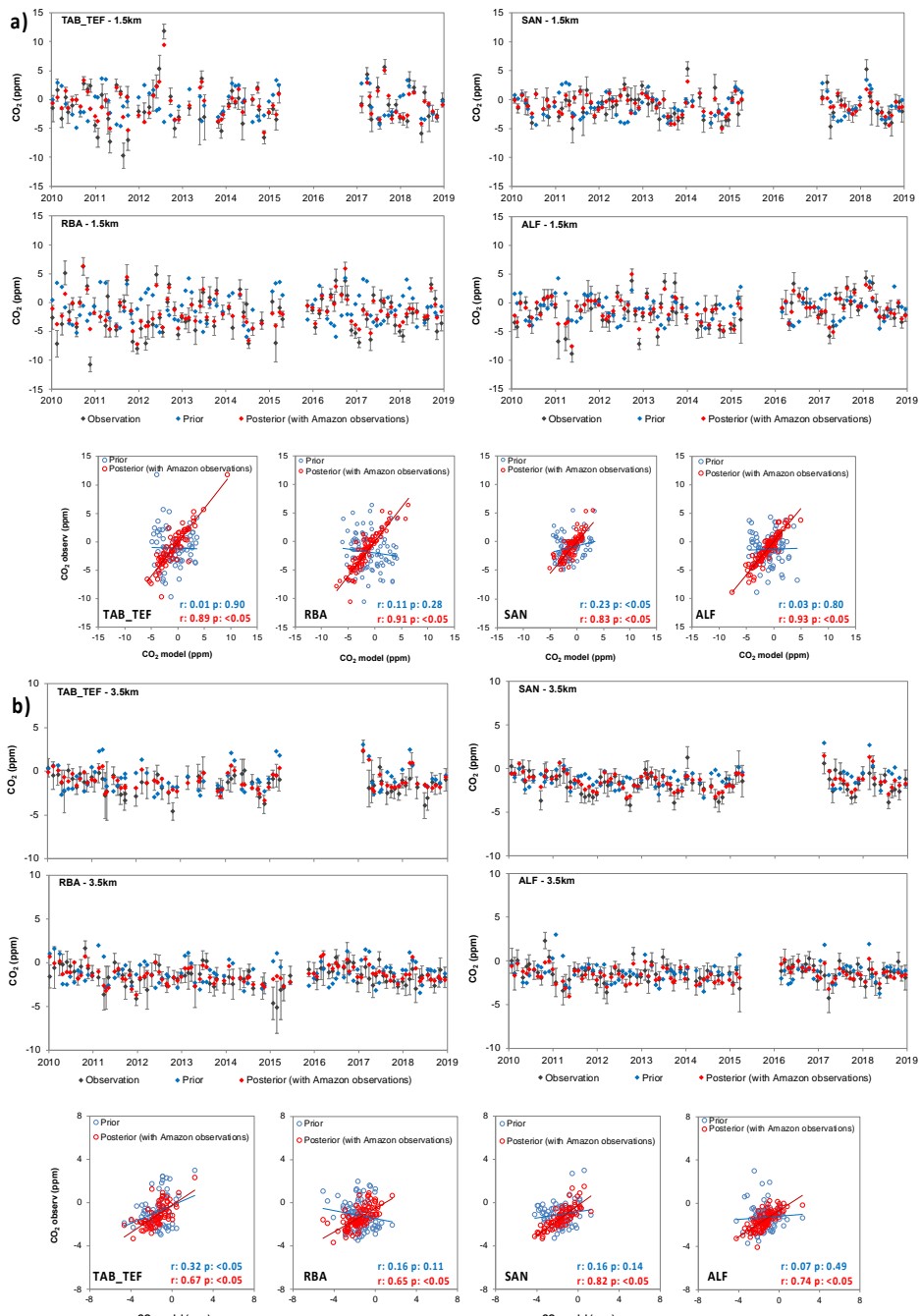

Figure 2: Detrended monthly mean CO₂ mole fractions (ppm) for prior (with CASA as land-biosphere prior flux), posterior and Amazon vertical profiles and its linear regressions, where a) is the mean below 1.5 km altitude (planetary boundary layer levels and b) the mean above 3.5 km altitude (vertical profile free troposphere), for each of the vertical profile sites. The model results were extracted for the grid cell where each site is located. After detrended we subtracted the global mean mole fraction from the observation and model mole fractions. Error bars represent the observation uncertainties.

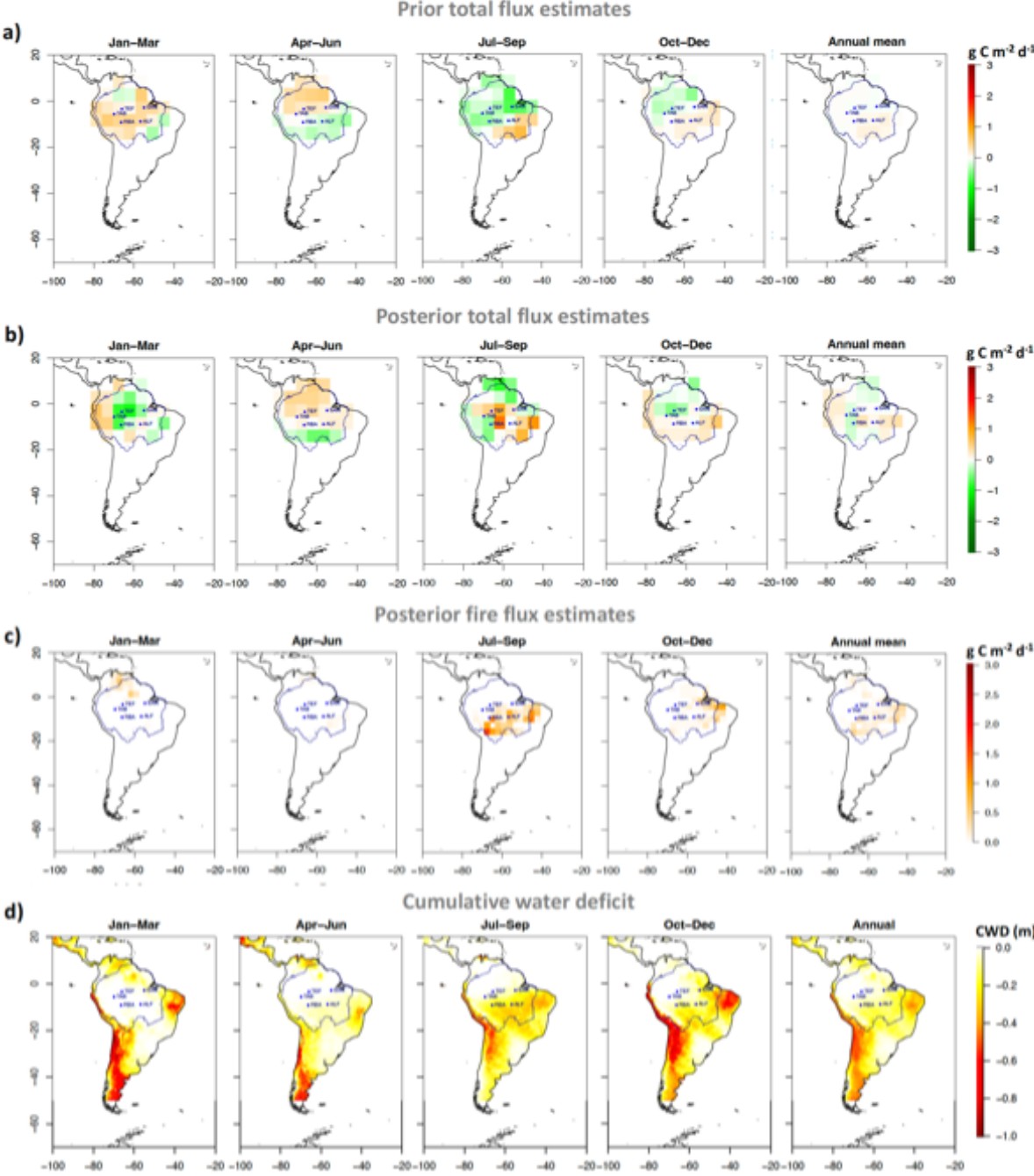


**Figure 3: Quarterly and annual mean a) prior total (with CASA as land-biosphere prior flux), b) posterior total, c) posterior fire carbon fluxes, where a positive value indicates a net emission of C while a negative value indicates a net uptake, d) cumulative water deficit (CWD) for the Amazon region between 2010 and 2018. The blue contour represents the Amazon area based on Eva and Huber (2005).**

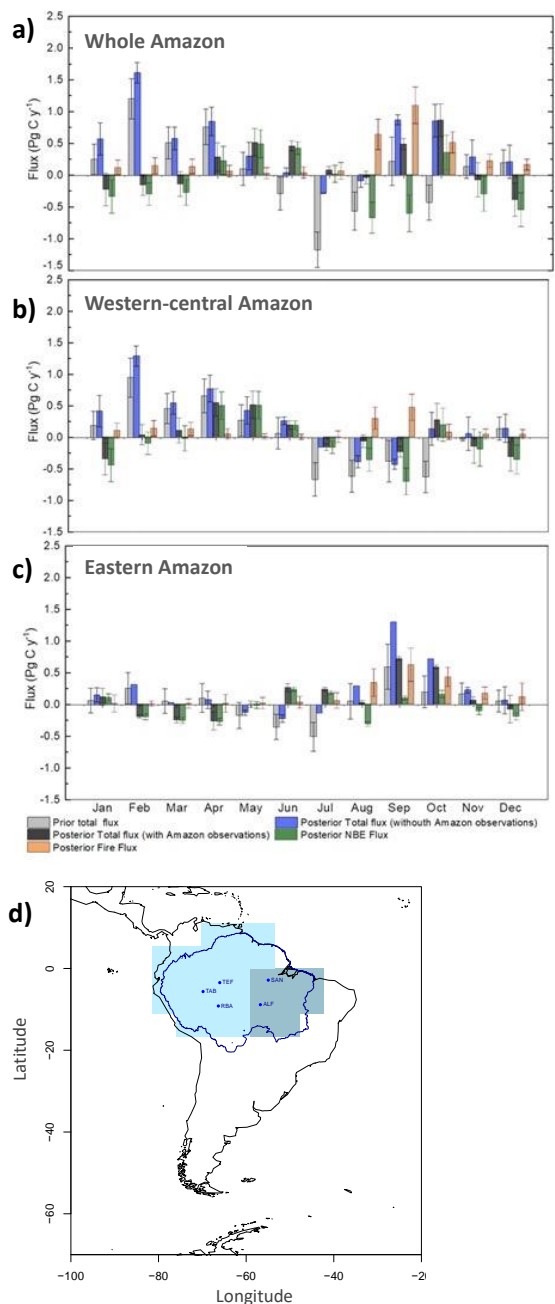


Figure 4: Nine-year monthly mean (2010-2018) carbon fluxes for the a) whole Amazon, b) western-central Amazon and c) eastern Amazon areas: prior total flux (grey bars), posterior total flux without the Amazon vertical profile observations in the inversion (blue bars), posterior total flux with the Amazon vertical profile observations in the inversion (black bars), posterior fire fluxes using MOPPIT carbon monoxide observations in the inversion (orange bars) and posterior NBE fluxes which is the result of the 1015 subtraction of the posterior fire fluxes from the posterior total fluxes the Amazon vertical profile observations in the inversion (green bars), representing the net biome exchange. Error bars represents the monthly mean uncertainties d) Amazon mask used in the study, the whole Amazon area is the sum of western-central Amazon and eastern Amazon areas. The blue contour represents the Amazon area based on Eva and Huber (2005).

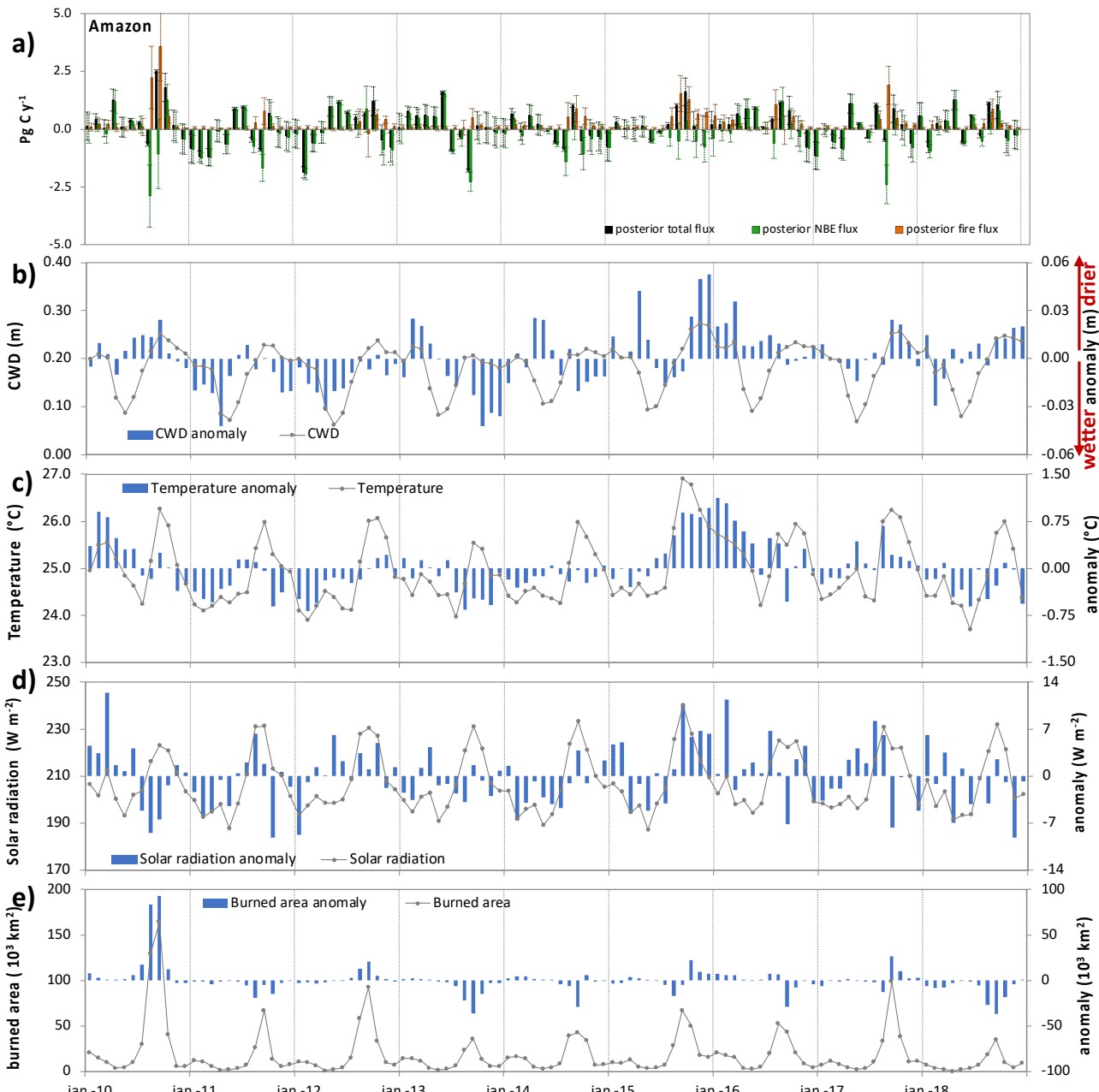

**Figure 5: a) Monthly mean carbon fluxes for the whole Amazon area: posterior total flux with the Amazon vertical profile observations in the inversion (black bars), posterior fire fluxes using MOPITT carbon monoxide observations in the inversion (orange bars) and posterior NBE fluxes which is the result of the subtraction of the posterior fire fluxes from the posterior total fluxes the Amazon vertical profile observations in the inversion (green bars), representing the net biome exchange. Monthly mean and anomalies of b) cumulative water deficit (CWD), c) temperature, d) shortwave flux down solar radiation (all sky) and e) burned area for the Amazon area.**


**Table 1: Nine-year mean prior total, posterior total without the vertical profile observations assimilated in the inversions, posterior total with the vertical profile observations assimilated in the inversions and fire fluxes for the whole Amazon, west-central and east Amazon regions.**

| Amazon C land fluxes 2010-2018 (Pg C y$^{-1}$) | | | |
|---|---|---|---|
| **Region** | **Amazon** | **West-central Amazon** | **East Amazon** |
| **Prior total flux** | $0.08 \pm 0.24$ | $0.03 \pm 0.21$ | $0.04 \pm 0.20$ |
| **Posterior total flux (without Amazon observations)** | $0.48 \pm 0.17$ | $0.26 \pm 0.16$ | $0.23 \pm 0.07$ |
| **Posterior total flux (with Amazon observations)** | $0.13 \pm 0.17$ | $0.03 \pm 0.17$ | $0.10 \pm 0.08$ |
| **Posterior fire flux** | $0.26 \pm 0.13$ | $0.11 \pm 0.10$ | $0.15 \pm 0.11$ |
| **Posterior NBE flux (without Amazon observations)** | $0.21 \pm 0.20$ | $0.12 \pm 0.18$ | $0.09 \pm 0.13$ |
| **Posterior NBE flux (with Amazon observations)** | $-0.13 \pm 0.20$ | $-0.08 \pm 0.18$ | $-0.05 \pm 0.13$ |

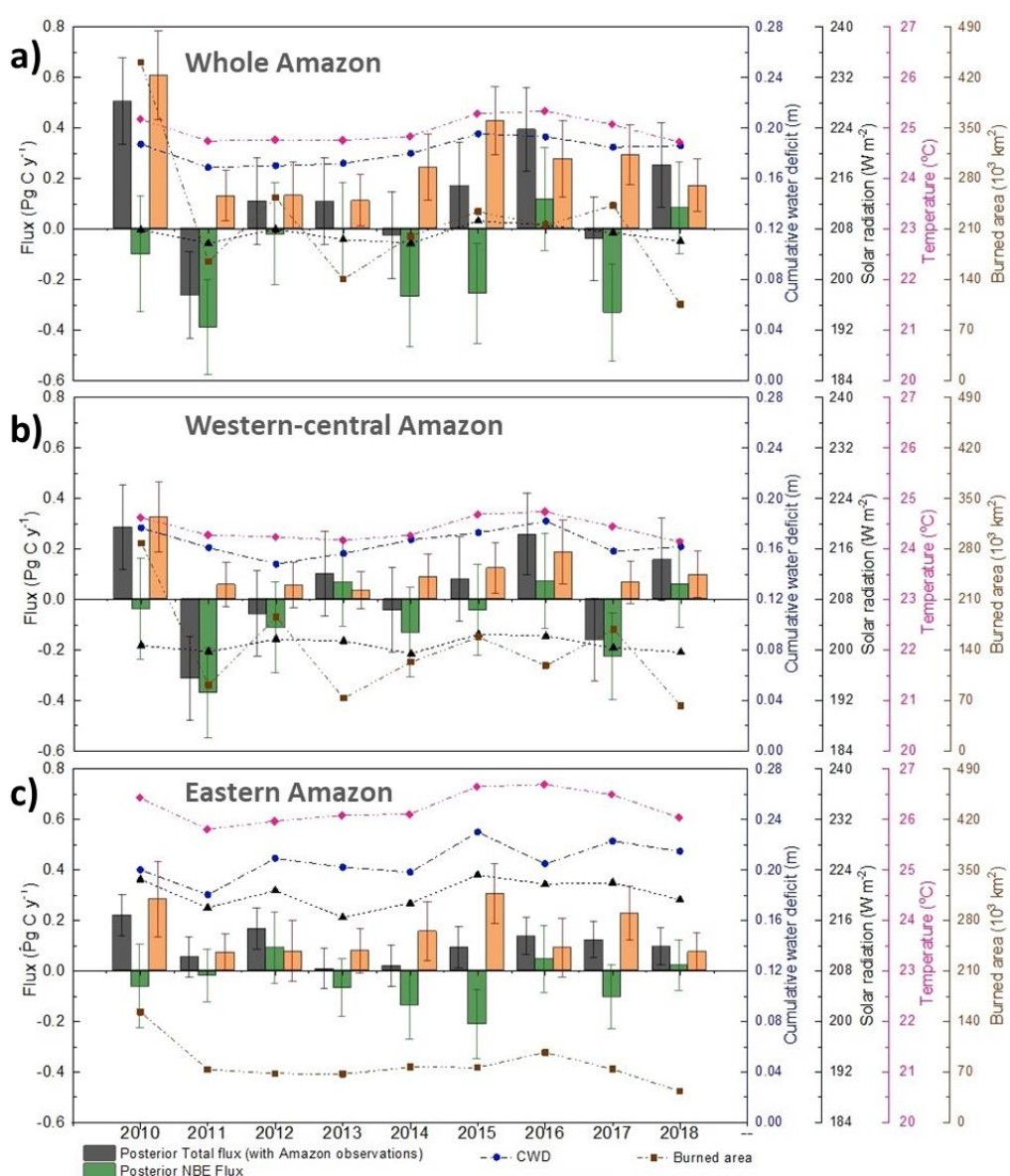

**Figure 6: Annual mean carbon fluxes for the a) whole Amazon, b) western-central and c) eastern Amazon areas: posterior total flux with the Amazon vertical profile observations in the inversion (black bars) and posterior fire fluxes using MOPITT carbon monoxide observations in the inversion (red bars). Annual cumulative water deficit (blue line), annual mean temperature (pink line), annual mean shortwave flux down solar radiation (all sky; black line) and annual total burned area (brown line).**

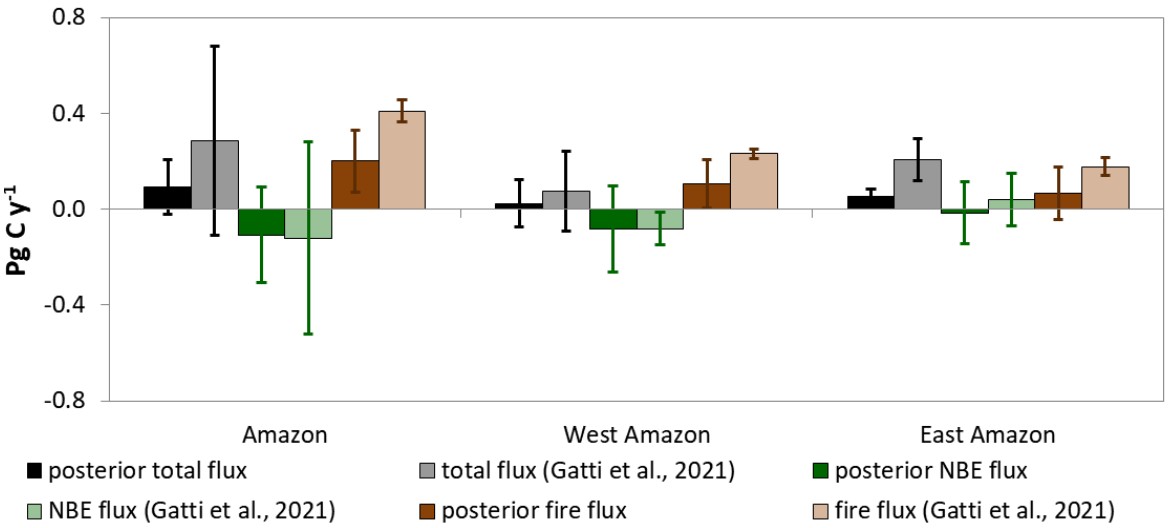

**Figure 7: Comparison of nine-year mean of carbon fluxes from the inverse modeling (prior total flux, posterior total flux, posterior NBE flux (total minus fire emissions) and posterior fire flux), and fluxes estimates (total, NBE and fire) using a mass balance technique in Gatti et al. (2021). All fluxes are estimated using the Amazon areas (km$^2$) from Gatti et al. (2021).**

**Appendix A**

Table A1. NOAA monitoring sites with $CO_2$ observations data used in the inverse model.

| Code | Name | Latitude | Longitude | Time-period |
|---|---|---|---|---|
| ALT | Alert, Nunavut, Canada | 82.45° N | 62.50° W | 2010-2018 |
| AMY | Anmyeon-do, Republic of Korea | 36.53° N | 126.32° E | 2013-2018 |
| ASC | Ascension Island, United Kingdom | 7.96° S | 14.40° W | 2010-2018 |
| ASK | Assekrem, Algeria | 23.26° N | 5.63° E | 2010-2018 |
| AZR | Terceira Island, Azores, Portugal | 38.76° N | 27.37° W | 2010-2018 |
| BAL | Baltic Sea, Poland | 55.35° N | 17.22° E | 2010-2011 |
| BHD | Baring Head Station, New Zealand | 41.40° S | 174.87° E | 2010-2018 |
| BKT | Bukit Kototabang, Indonesia | 0.20° S | 100.31° E | 2010-2018 |
| BMW | Tudor Hill, Bermuda, United Kingdom | 32.26° N | 64.87° W | 2010-2018 |
| BRW | Barrow Atmospheric Baseline Observatory, United States | 71.32° N | 156.61° W | 2010-2018 |
| CBA | Cold Bay, Alaska, United States | 55.21° N | 162.72° W | 2010-2018 |
| CGO | Cape Grim, Tasmania, Australia | 40.68° S | 144.69° E | 2010-2018 |
| CHR | Christmas Island, Republic of Kiribati | 1.70° N | 157.15° W | 2010-2018 |
| CIB | Centro de Investigacion de la Baja Atmosfera (CIBA), Spain | 41.81° N | 4.93° W | 2010-2018 |
| CPT | Cape Point, South Africa | 34.35° S | 18.48° E | 2012-2018 |
| CRZ | Crozet Island, France | 46.43° S | 51.84° E | 2010-2018 |
| DRP | Drake Passage, N/A | 59.00° S | 64.69° W | 2010-2018 |
| DSI | Dongsha Island, Taiwan | 20.69° N | 116.72° E | 2010-2018 |

| | | | | |
|---|---|---|---|---|
| EIC | Easter Island, Chile | 27.15° S | 109.42° W | 2010-2018 |
| GMI | Mariana Islands, Guam | 13.38° N | 144.65° E | 2010-2018 |
| HBA | Halley Station, Antarctica, United Kingdom | 75.605° S | 26.21° W | 2010-2018 |
| HPB | Hohenpeissenberg, Germany | 47.80° N | 11.02° E | 2010-2018 |
| HSU | Humboldt State University, United States | 41.05° N | 124.75° W | 2010-2017 |
| HUN | Hegyhatsal, Hungary | 46.95° N | 16.65° W | 2010-2018 |
| ICE | Storhofdi, Vestmannaeyjar, Iceland | 63.39° N | 20.28° W | 2016-2018 |
| IZO | Izana, Tenerife, Canary Islands, Spain | 28.30° N | 16.49° W | 2010-2018 |
| KEY | Key Biscayne, Florida, United States | 25.66° N | 80.15° W | 2010-2018 |
| KUM | Cape Kumukahi, Hawaii, United States | 19.73° N | 155.01° W | 2010-2018 |
| LLB | Lac La Biche, Alberta, Canada | 54.95° N | 112.46° W | 2010-2013 |
| LLN | Lulin, Taiwan | 23.47° N | 120.87° E | 2010-2018 |
| LMP | Lampedusa, Italy | 35.51° N | 12.63° E | 2010-2018 |
| MEX | High Altitude Global Climate Observation Center, Mexico | 18.98° N | 97.31° W | 2010-2018 |
| MHD | Mace Head, County Galway, Ireland | 53.32° N | 9.89° W | 2010-2018 |
| MID | Sand Island, Midway, United States | 28.21° N | 177.38° W | 2010-2018 |
| MLO | Mauna Loa, Hawaii, United States | 19.53° N | 155.57° W | 2010-2018 |
| NAT | Farol De Mae Luiza Lighthouse, Brazil | 5.79° S | 35.18° W | 2010-2018 |
| NMB | Gobabeb, Namibia | 23.58° S | 15.03° E | 2010-2018 |
| NWR | Niwot Ridge, Colorado, United States | 40.05° N | 105.58° W | 2010-2018 |
| OXK | Ochsenkopf, Germany | 50.03° N | 11.80° E | 2010-2018 |
| PAL | Pallas-Sammaltunturi, GAW Station, Finland | 67.97° N | 24.11° E | 2010-2018 |
| POC000 | Pacific Ocean (0° N) | 0.00° | - | 2010-2017 |
| POCN05 | Pacific Ocean (5° N) | 5.00° N | - | 2010-2017 |
| POCN10 | Pacific Ocean (10° N) | 10.00° N | - | 2010-2017 |
| POCN15 | Pacific Ocean (15° N) | 15.00° N | - | 2010-2017 |
| POCN20 | Pacific Ocean (20° N) | 20.00° N | - | 2010-2017 |
| POCN25 | Pacific Ocean (25° N) | 25.00° N | - | 2010-2017 |
| POCN30 | Pacific Ocean (30° N) | 30.00° N | - | 2010-2017 |
| POCS05 | Pacific Ocean (5° S) | 5.00° S | - | 2010-2017 |
| POCS10 | Pacific Ocean (10° S) | 10.00° S | - | 2010-2017 |
| POCS15 | Pacific Ocean (15° S) | 15.00° S | - | 2010-2017 |
| POCS20 | Pacific Ocean (20° S) | 20.00° S | - | 2010-2017 |
| POCS25 | Pacific Ocean (25° S) | 25.00° S | - | 2010-2017 |
| POCS30 | Pacific Ocean (30° S) | 30.00° S | - | 2010-2017 |
| PSA | Palmer Station, Antarctica, United States | 64.77° S | 64.05° W | 2010-2018 |
| PTA | Point Arena, California, United States | 38.95° N | 123.74° W | 2010-2011 |
| RPB | Ragged Point, Barbados | 13.16° N | 59.43° W | 2010-2018 |
| SDZ | Shangdianzi, China | 40.65° N | 117.11° E | 2010-2015 |
| SEY | Mahe Island, Seychelles | 4.68° S | 55.53° E | 2010-2018 |

| | | | | |
|---|---|---|---|---|
| SGP | Southern Great Plains, Oklahoma, United States | 36.60° N | 97.48° W | 2010-2018 |
| SHM | Shemya Island, Alaska, United States | 52.71° N | 174.12° E | 2010-2018 |
| SMO | Tutuila, American Samoa | 14.24° S | 170.56° W | 2010-2018 |
| SUM | Summit, Greenland | 72.59° N | 38.42° W | 2010-2018 |
| SYO | Syowa Station, Antarctica, Japan | 69.01° S | 39.59° E | 2010-2018 |
| TAP | Tae-ahn Peninsula, Republic of Korea | 36.73° N | 126.13° E | 2010-2018 |
| THD | Trinidad Head, California, United States | 41.05° N | 124.15° W | 2010-2017 |
| TIK | Hydrometeorological Observatory of Tiksi, Russia | 71.59° N | 128.88° E | 2011-2018 |
| USH | Ushuaia, Argentina | 54.84° S | 68.31° W | 2010-2018 |
| UTA | Wendover, Utah, United States | 39.90° N | 113.71° W | 2010-2018 |
| UUM | Ulaan Uul, Mongolia | 44.45° N | 111.09° E | 2010-2018 |
| WIS | Weizmann Institute of Science at the Arava Institute, Ketura, Israel | 29.96° N | 35.06° E | 2010-2018 |
| WLG | Mt. Waliguan, Peoples Republic of China | 36.28° N | 100.89° E | 2010-2018 |
| ZEP | Ny-Alesund, Svalbard, Norway and Sweden | 78.90° N | 11.88° E | 2010-2018 |

Table A2. Mean difference between $CO_2$ mole fraction model estimates and observations.

**$CO_2$ mole fraction mean difference (ppm)**

**CASA as land-biosphere prior flux**

| Site | Mean below 1.5 km altitude | | Mean above 3.5 km altitude | |
|---|---|---|---|---|
| | Prior - observed | Posterior - observed | Prior - observed | Posterior - observed |
| ALF | 3.0 | 1.3 | 1.2 | 0.7 |
| SAN | 2.3 | 1.3 | 1.3 | 0.6 |
| RBA | 4.1 | 1.3 | 1.5 | 0.7 |
| TAB_TEF | 3.5 | 1.4 | 1.4 | 0.7 |

**CARDAMOM as land-biosphere prior flux**

| Site | Mean below 1.5 km altitude | | Mean above 3.5 km altitude | |
|---|---|---|---|---|
| | Prior - observed | Posterior - observed | Prior - observed | Posterior - observed |
| ALF | -3.3 | 0.1 | -2.1 | -0.2 |
| SAN | -2.9 | 0.0 | -0.8 | 0.3 |
| RBA | -5.9 | -0.3 | -2.5 | 0.1 |
| TAB_TEF | -6.1 | 0.0 | -1.5 | 0.3 |


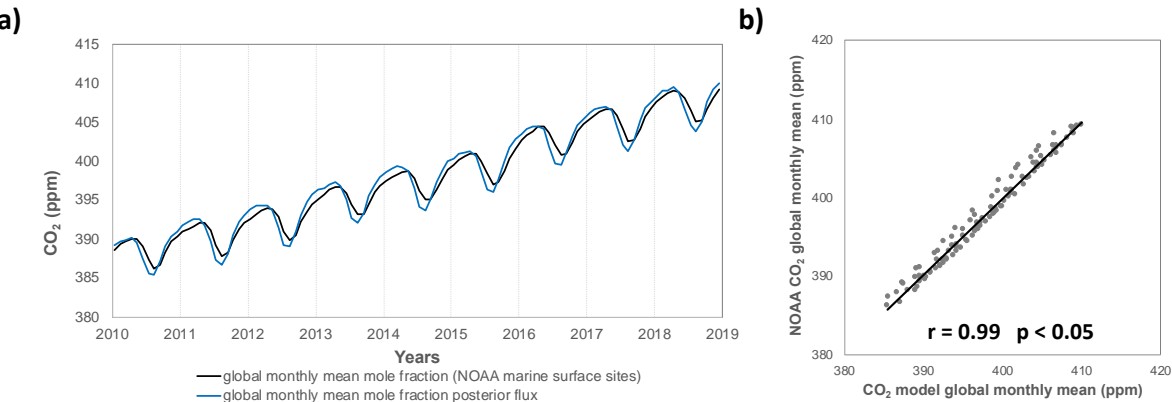

Figure A1. a) Time series of the global monthly mean $CO_2$ mole fraction from our posteriori fluxes and from the NOAA marine surface sites (Lans, et al., 2023). b) Linear correlation between global monthly mean $CO_2$ mole fraction from our posteriori fluxes and from the NOAA marine surface sites.

**Posterior total flux estimates without assimilated Amazon vertical profiles**

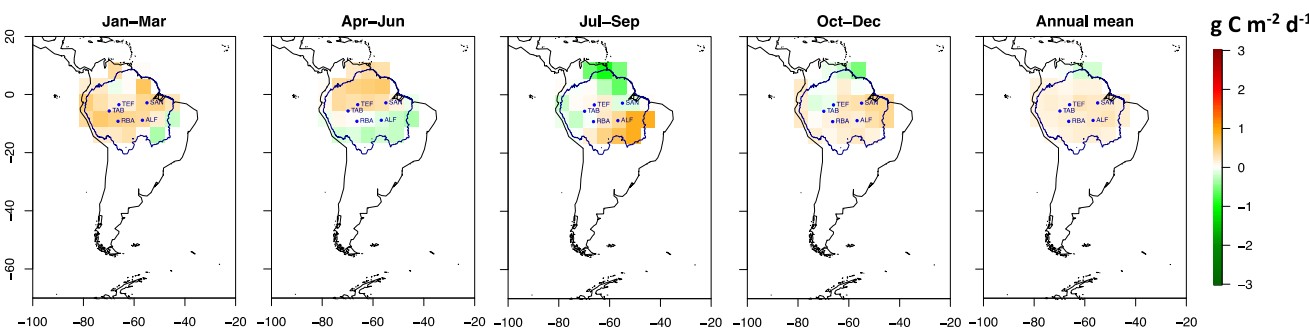

Figure A2. Quarterly and annual mean posterior total carbon fluxes without assimilated Amazon vertical profile data for the Amazon region between 2010 and 2018. The blue contour represents the Amazon area based on Eva and Huber (2005).

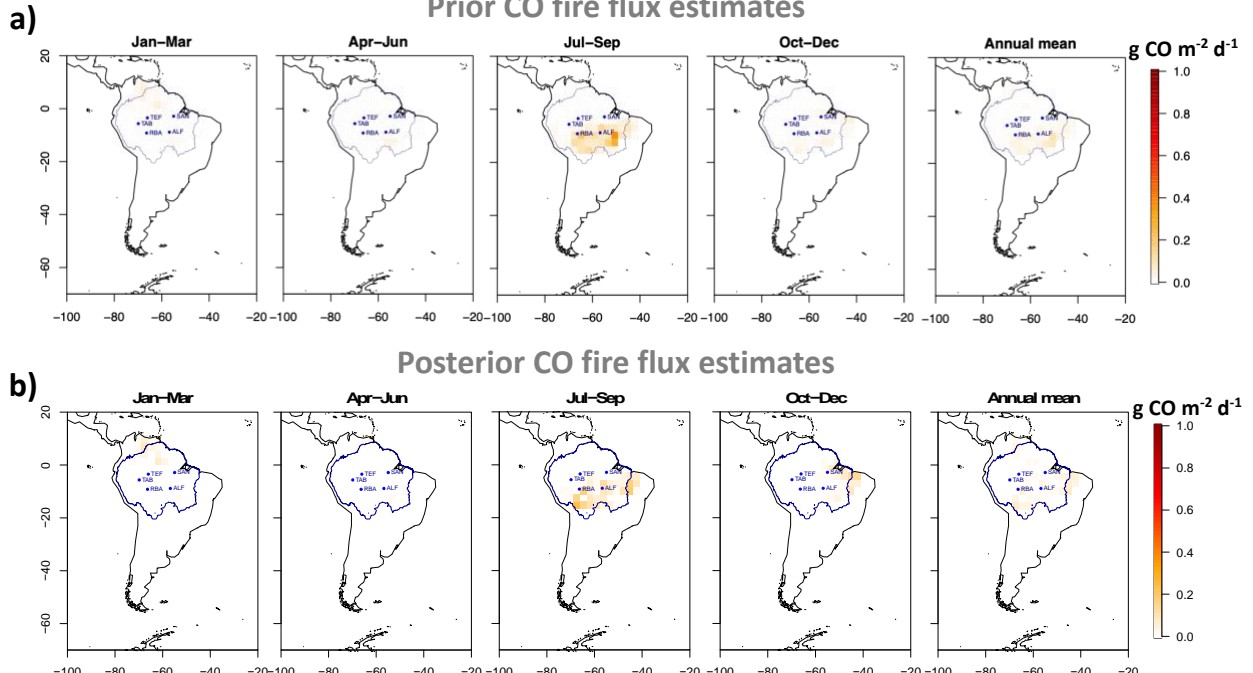

Figure A3. Quarterly and annual mean prior (a) and posterior (b) fire carbon monoxide (CO) fluxes with
MOPITT data assimilated for the Amazon region between 2010 and 2018. The blue contour represents
the Amazon area based on Eva and Huber (2005).

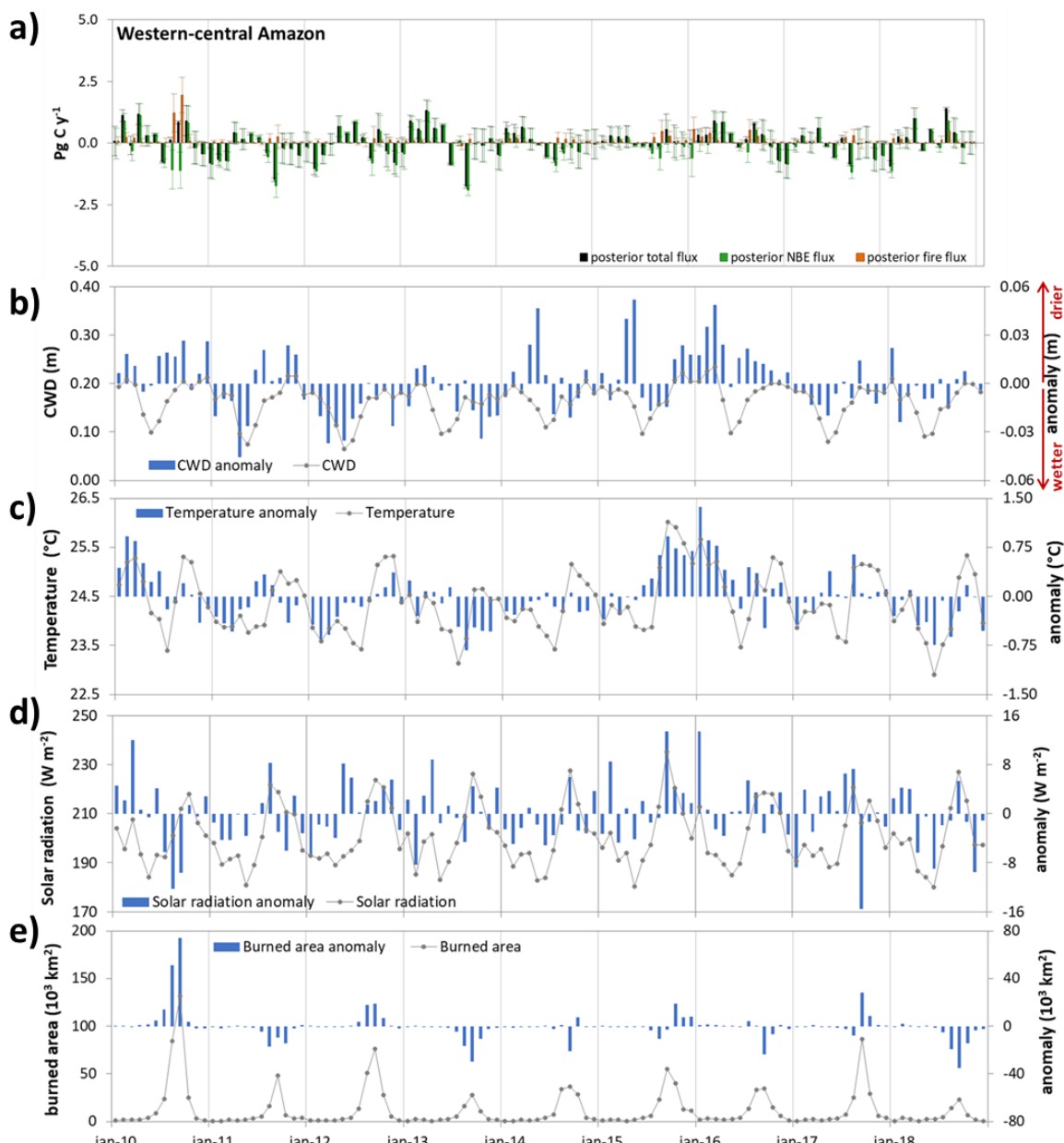


Figure A4. a) Monthly mean carbon fluxes for the western-central Amazon area: posterior total flux with the Amazon vertical profile observations in the inversion (black bars), posterior fire fluxes using MOPPIT carbon monoxide observations in the inversion (orange bars) and posterior NBE fluxes which is the result of the subtraction of the posterior fire fluxes from the posterior total fluxes the Amazon vertical profile

observations in the inversion (green bars), representing the net biome exchange. Monthly mean and anomalies of b) cumulative water deficit (CWD), c) temperature, d) shortwave flux down solar radiation (all sky) and e) burned area for the western-central Amazon area.

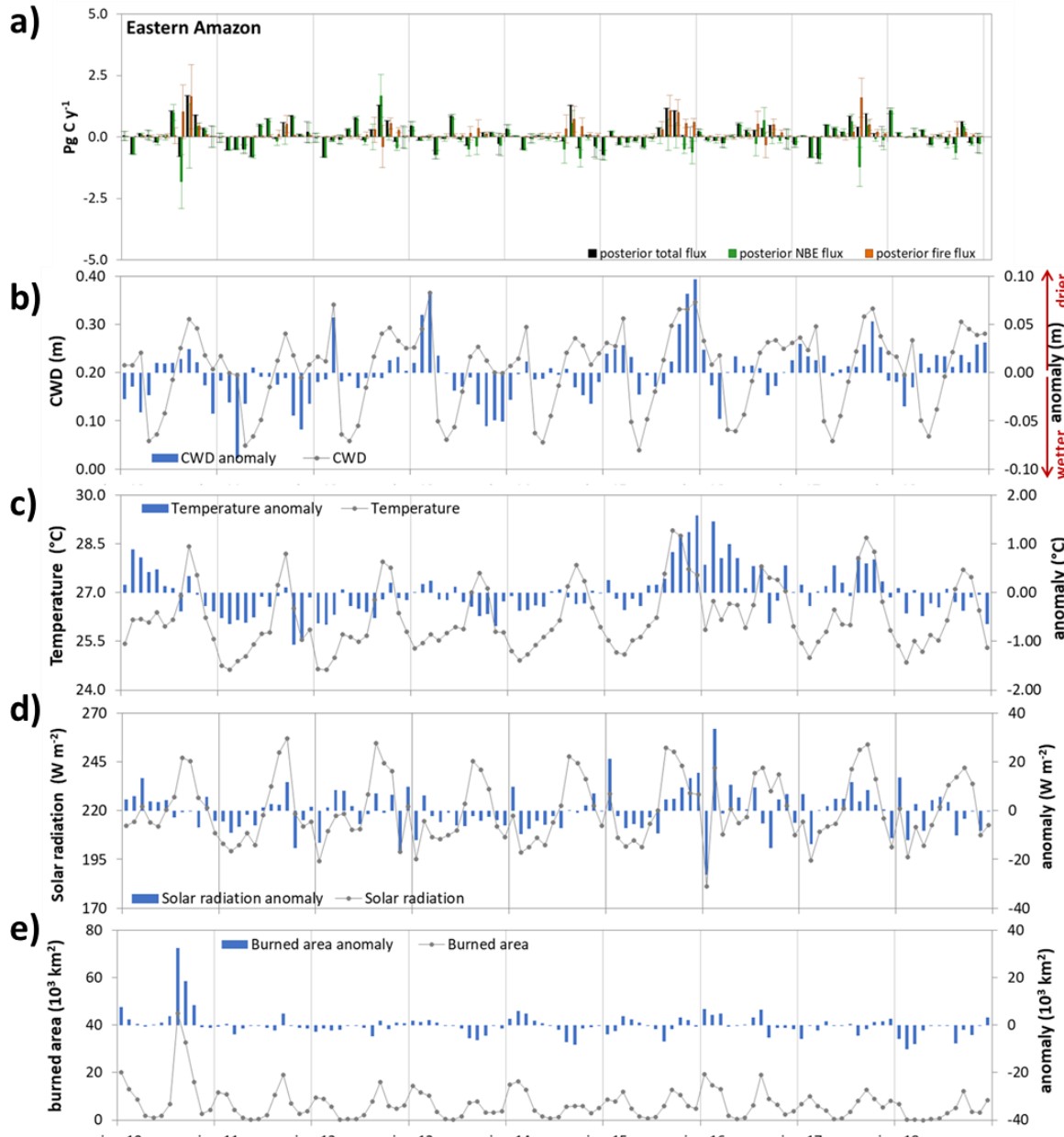

Figure A5. a) Monthly mean carbon fluxes for the eastern Amazon area: posterior total flux with the
Amazon vertical profile observations in the inversion (black bars), posterior fire fluxes using MOPPIT
carbon monoxide observations in the inversion (orange bars) and posterior NBE fluxes which is the result
of the subtraction of the posterior fire fluxes from the posterior total fluxes the Amazon vertical profile
observations in the inversion (green bars), representing the net biome exchange. Monthly mean and
anomalies of b) cumulative water deficit (CWD), c) temperature, d) shortwave flux down solar radiation
(all sky) and e) burned area for the eastern Amazon area.

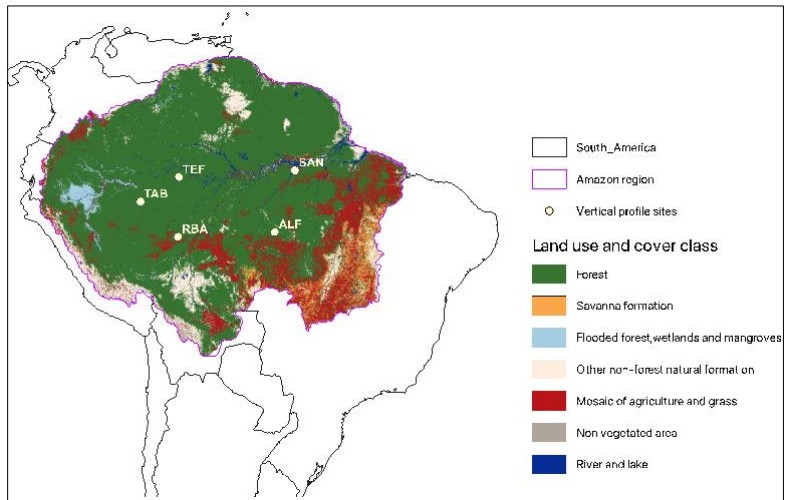

Figure A6. Map of land use and cover data from Mapbiomas (2020) for Pan-Amazonia up to 2018. Purple line represents the Amazon region boundaries (based on Eva and Huber, 2005) and grey line the South America boundaries.


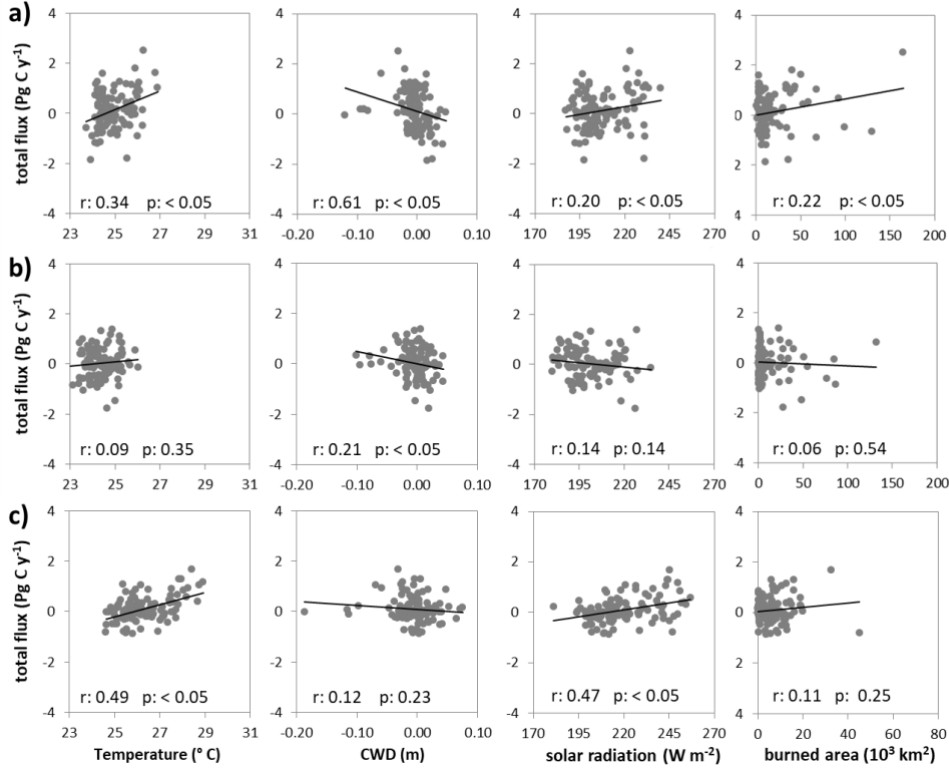

Figure A7. a) Linear regressions between monthly mean carbon posterior total flux and temperature, cumulative water deficit (CWD), solar radiation and burned area for a) whole, b) western-central and c) eastern Amazon regions.

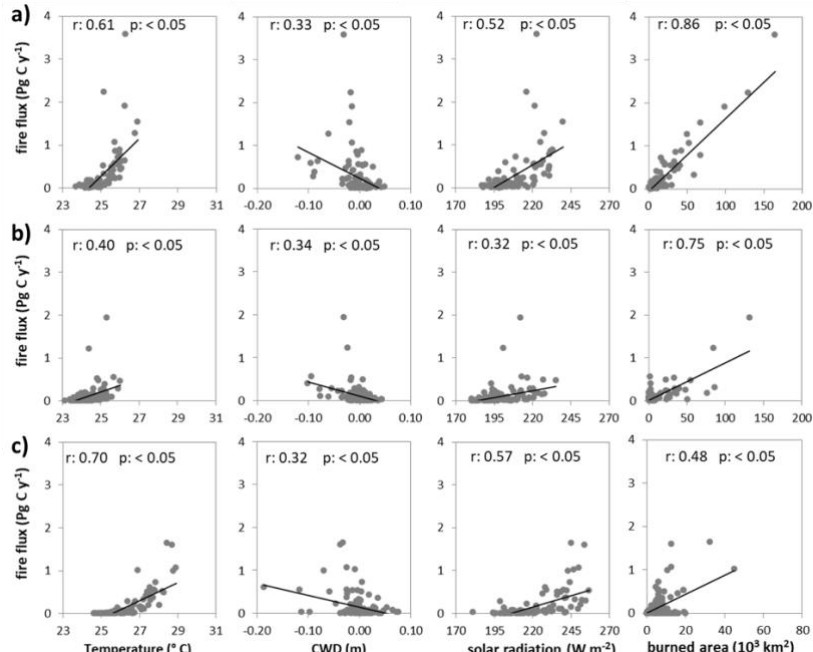

Figure A8. a) Linear regressions between monthly mean carbon posterior fire flux and temperature, cumulative water deficit (CWD), solar radiation and burned area for a) whole, b) western-central and c) eastern Amazon regions.

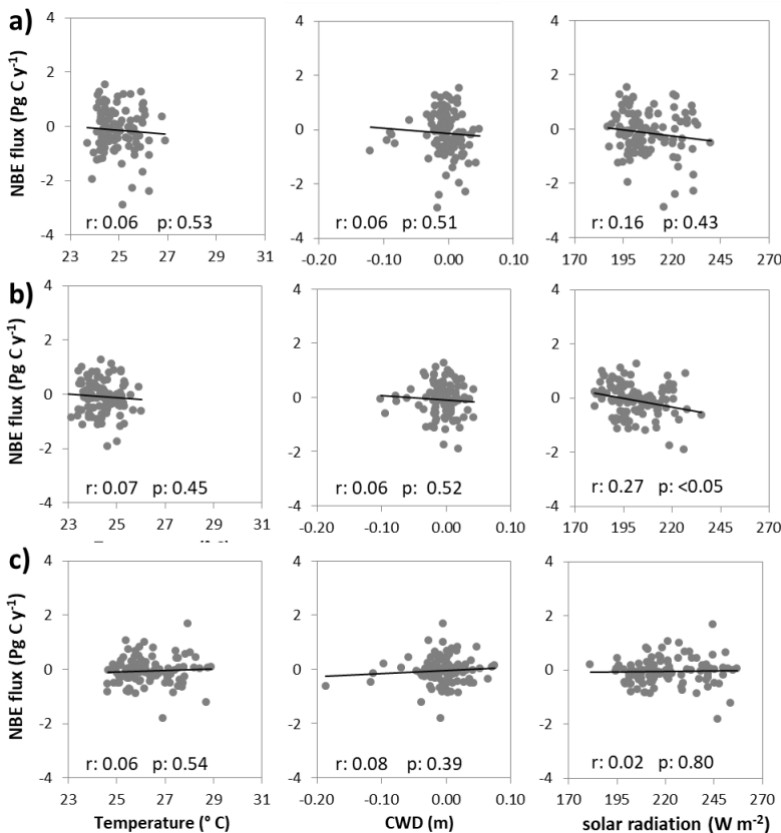

Figure A9. a) Linear regressions between monthly mean carbon posterior NBE flux (posterior total flux less posterior fire flux) and temperature, cumulative water deficit (CWD) and solar radiation for a) whole, b) western-central and c) eastern Amazon regions.


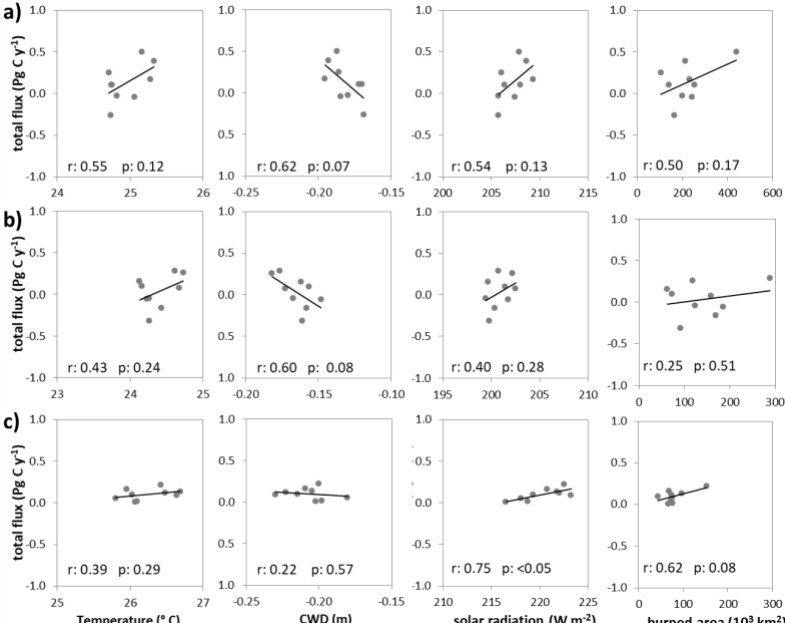

Figure A10. a) Linear regressions between annual mean carbon posterior total flux (posterior total flux less posterior fire flux) and temperature, cumulative water deficit (CWD), solar radiation and burned area for a) whole, b) western-central and c) eastern Amazon regions.


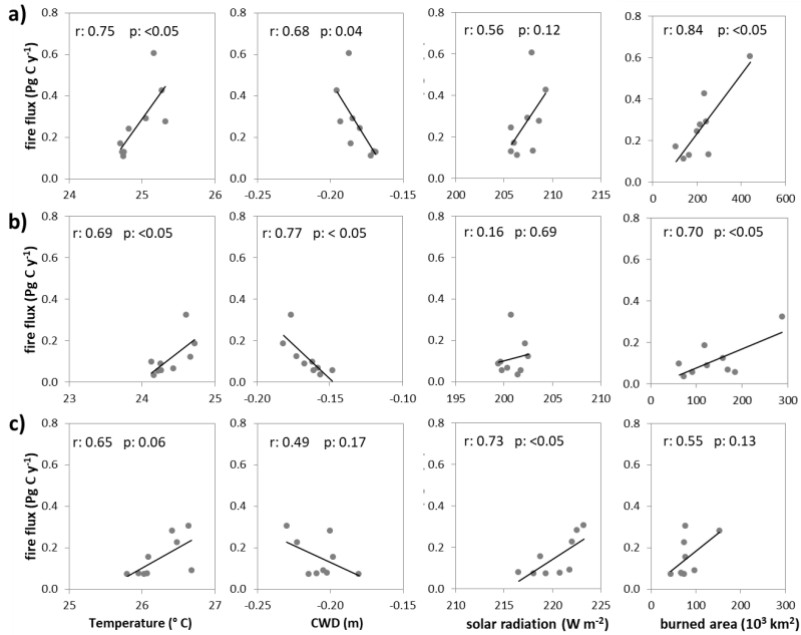

Figure A11. a) Linear regressions between annual mean carbon posterior fire flux (posterior total flux less posterior fire flux) and temperature, cumulative water deficit (CWD), solar radiation and burned area for a) whole, b) western-central and c) eastern Amazon regions.

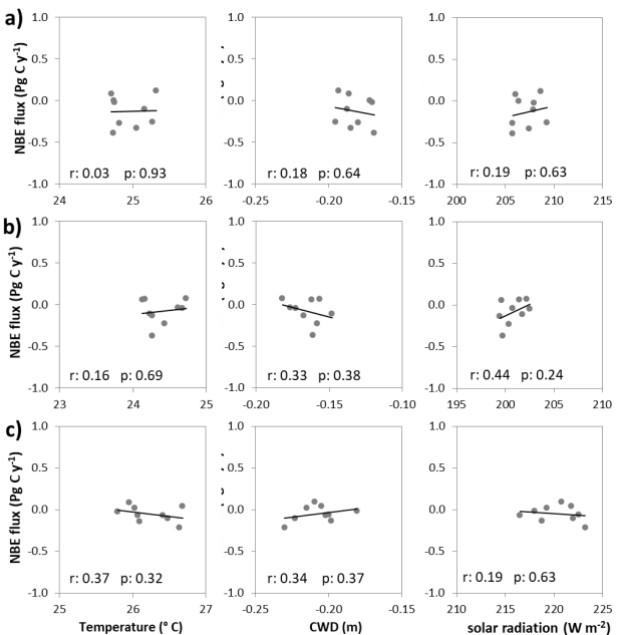


Figure A12. a) Linear regressions between annual mean carbon posterior NBE flux (posterior total flux less posterior fire flux) and temperature, cumulative water deficit (CWD), and solar radiation for a) whole, b) western-central and c) eastern Amazon regions.

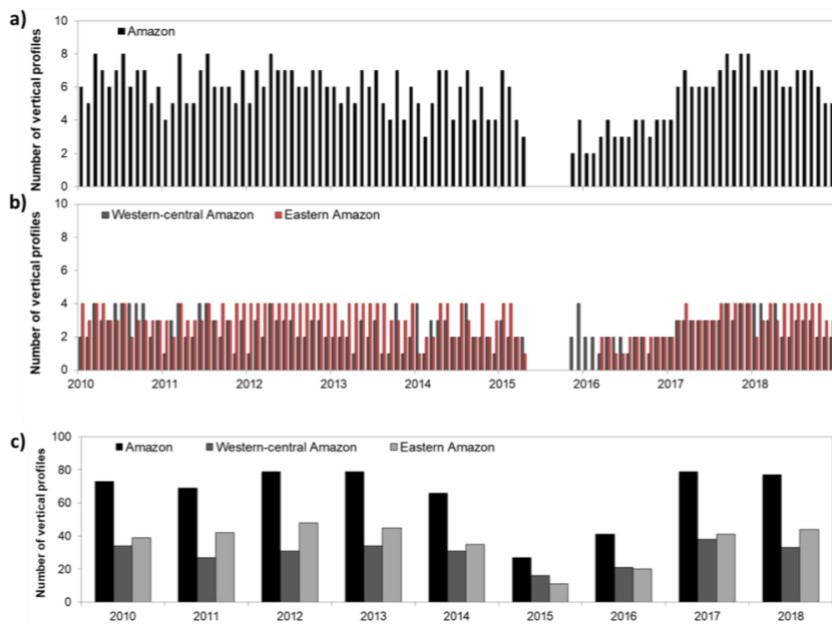


Figure A13. Total number of vertical profiles by month used in the inversions for the a) whole Amazon area, and b) divided in western-central (dark grey bars) and eastern Amazon regions (red bars). c) Total number of vertical profiles for whole (black bars), western-central (dark grey bars) and eastern Amazon regions (light grey bars). All the vertical profile data used were from Gatti et al., 2021.


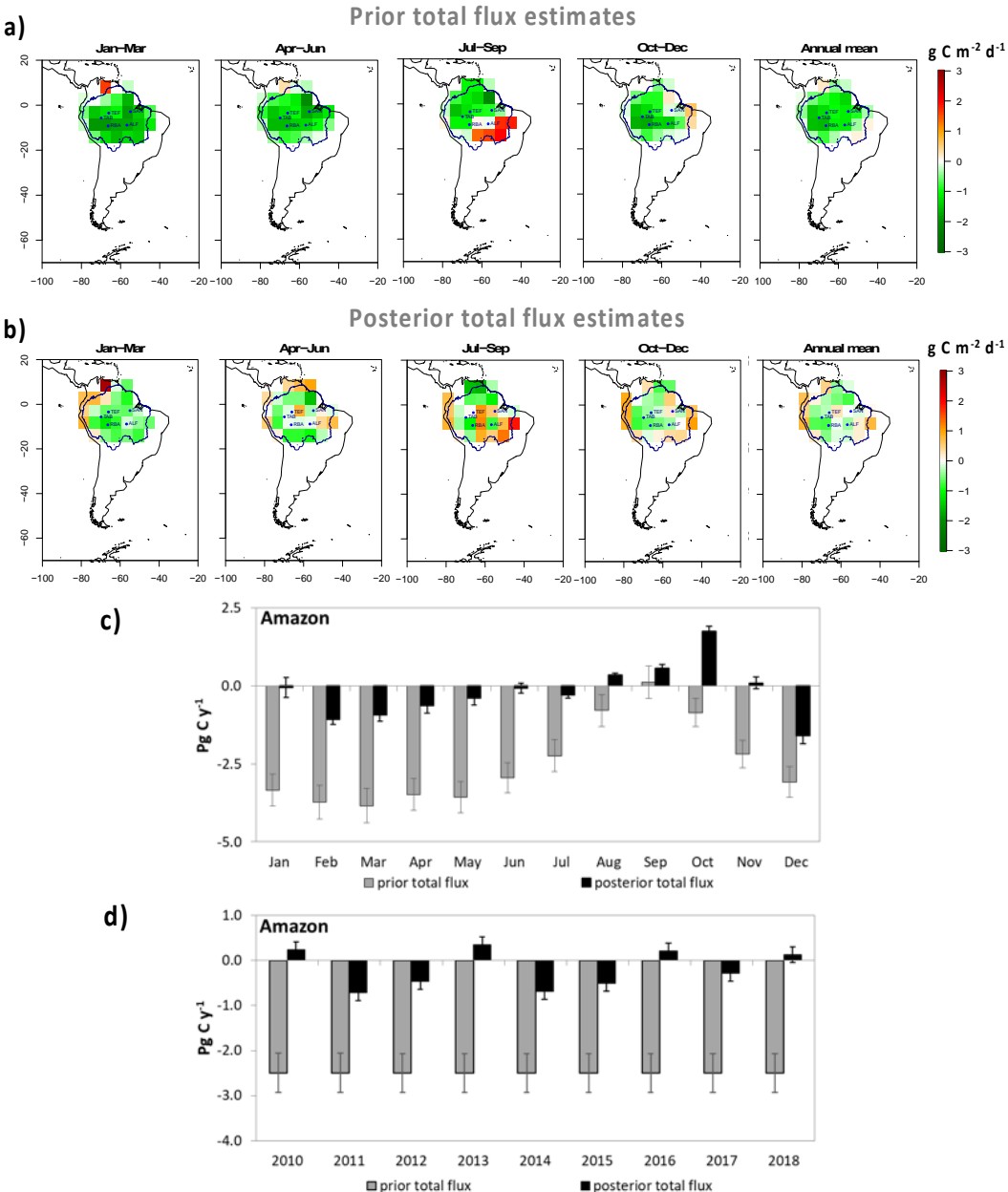

Figure A14. Quarterly and annual mean a) prior total (with CARDAMOM as land-biosphere prior flux), b) posterior total (with CARDAMOM as land-biosphere prior flux), carbon fluxes, where a positive value indicates a net emission of C while a negative value indicates a net uptake, c) nine-year monthly mean and d) annual means carbon fluxes for the Amazon using CARDAMOM estimates as land-biosphere prior fluxes between 2010 and 2018. The blue contour represents the Amazon area based on Eva and Huber (2005).

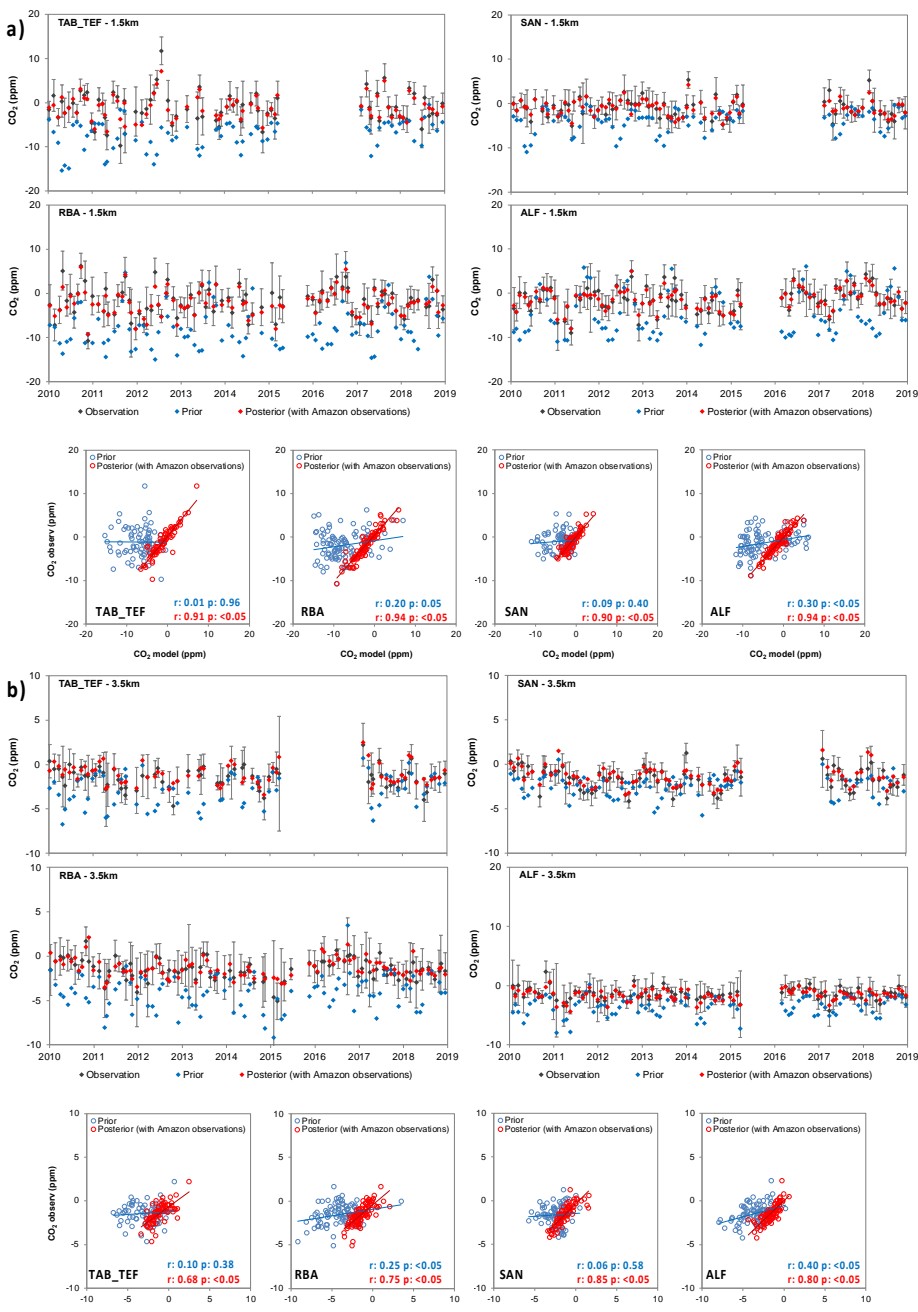

Figure A15: Detrended monthly mean $CO_2$ mole fractions (ppm) for prior (with CARDAMOM as land-biosphere prior flux), posterior and Amazon vertical profiles and its linear regressions, where a) is the mean below 1.5 km altitude (planetary boundary layer levels and b) the mean above 3.5 km altitude (vertical profile free troposphere), for each of the vertical profile sites. The model results were extracted for the grid cell where each site is located. After detrended we subtracted the global mean mole fraction from the observation and model mole fractions. Error bars represent the observation uncertainties.

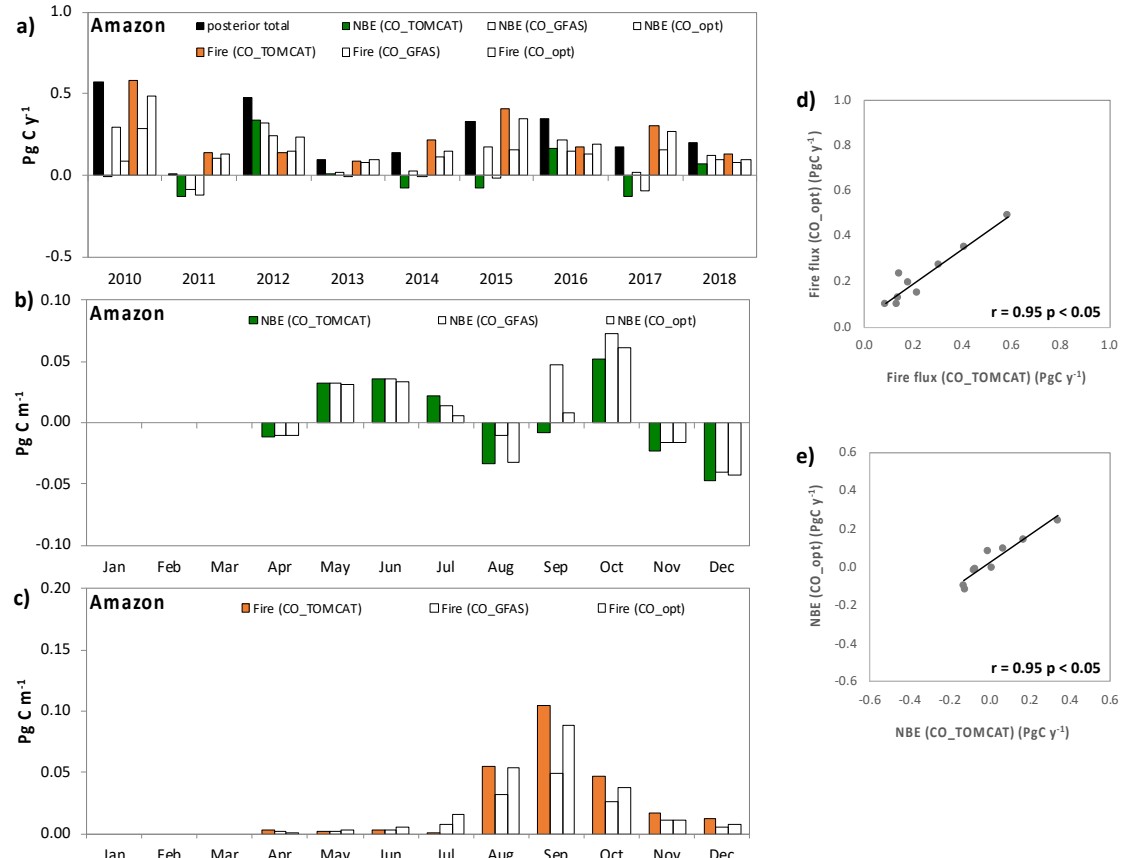

Figure A16. a) Annual mean fluxes for the Amazon region total, fire and NBE estimates. Fire and NBE based on TOMCAT CO inversions (CO_TOMCAT), Naus et al. (2022) emissions using GFAS as a prior (CO_GFAS) and with their CO optimized inversions (CO_opt). Nine-year monthly mean NBE (b) and fire (c) carbon fluxes for the Amazon, Fire and NBE based on TOMCAT CO inversions (CO_TOMCAT), Naus et al. (2022) emissions using GFAS as prior (CO_GFAS) and with their CO optimized inversions (CO_opt). Linear regressions between annual mean carbon fire flux (d) and posterior NBE (e) based on TOMCAT CO inversions (CO_TOMCAT) and Naus et al. (2022) CO optimized inversions (CO_opt)

Table A3. Annual mean fluxes (between April to December over the nine-year period, 2010 to 2018) using different CO estimates to estimate $CO_2$ fire and NBE fluxes.

| Carbon fluxes* (PgC $y^{-1}$) | | |
|---|---|---|
| Flux | NBE | Fire |
| CO_TOMCAT | 0.02 | 0.24 |
| CO_GFAS (Naus et al., 2022) | 0.12 | 0.14 |
| CO_opt (Naus et al., 2022) | 0.04 | 0.22 |

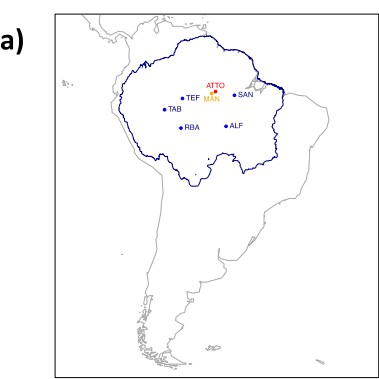

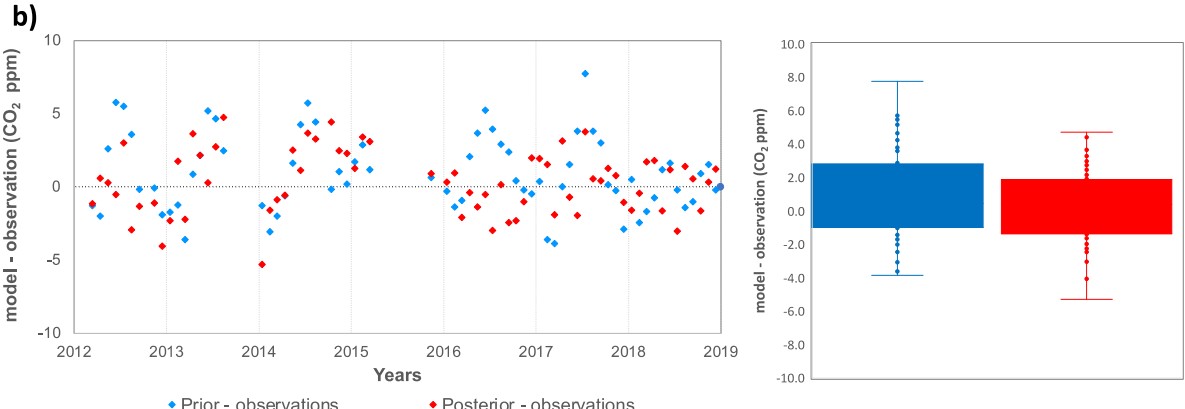


Figure A17. A) Locations of INPE/LaGEE Amazon vertical profile sites assimilated in the inversions (blue circles) and the two sites (MAN and ATTO) with data used for cross-validation. The blue contour represents the Amazon area based on Eva and Huber (2005). B) time series and box plot of model bias (model – observations) of monthly mean $CO_2$ mole fractions (ppm) for the ATTO tower.


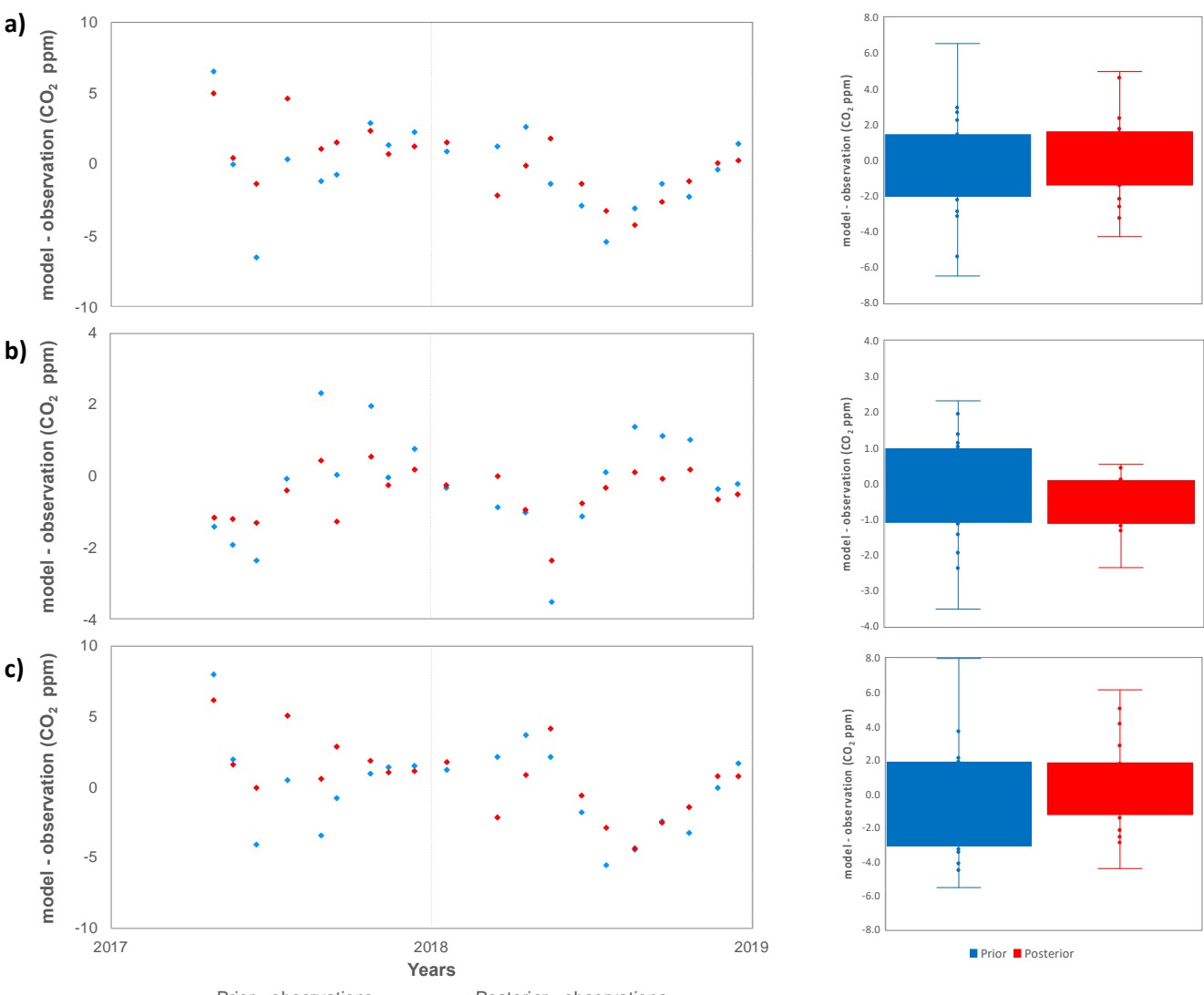

Figure A18. Time series and box plot of model bias (model – observations) of monthly mean $CO_2$ mole fractions (ppm) for the MAN vertical profiles (a) mean below 1.5km, (b) mean above 3.5km and the (c) difference between mean below 1.5km and mean above 3.5km.


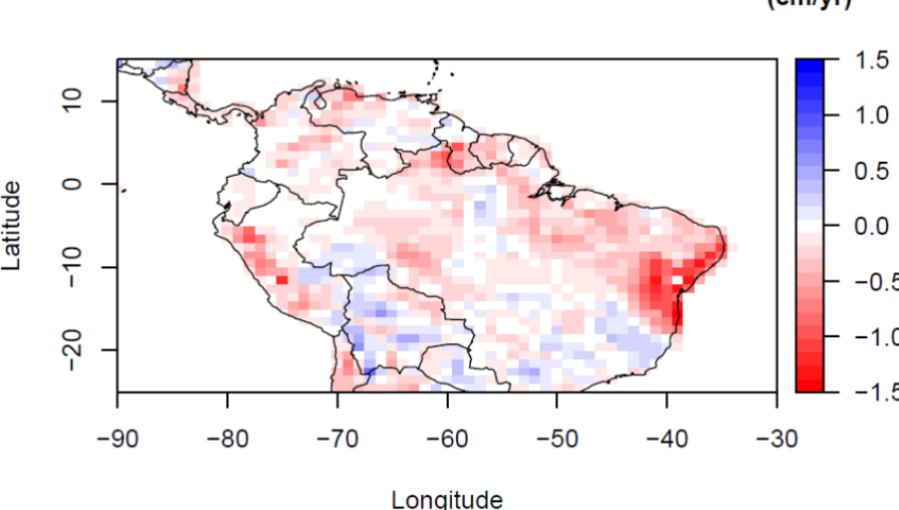

Figure A19. Time trend of maximum cumulative water deficit (CWD) between 1998 and 2019 based on TRMM v 7 precipitation estimates (Huffman et al., 2010).


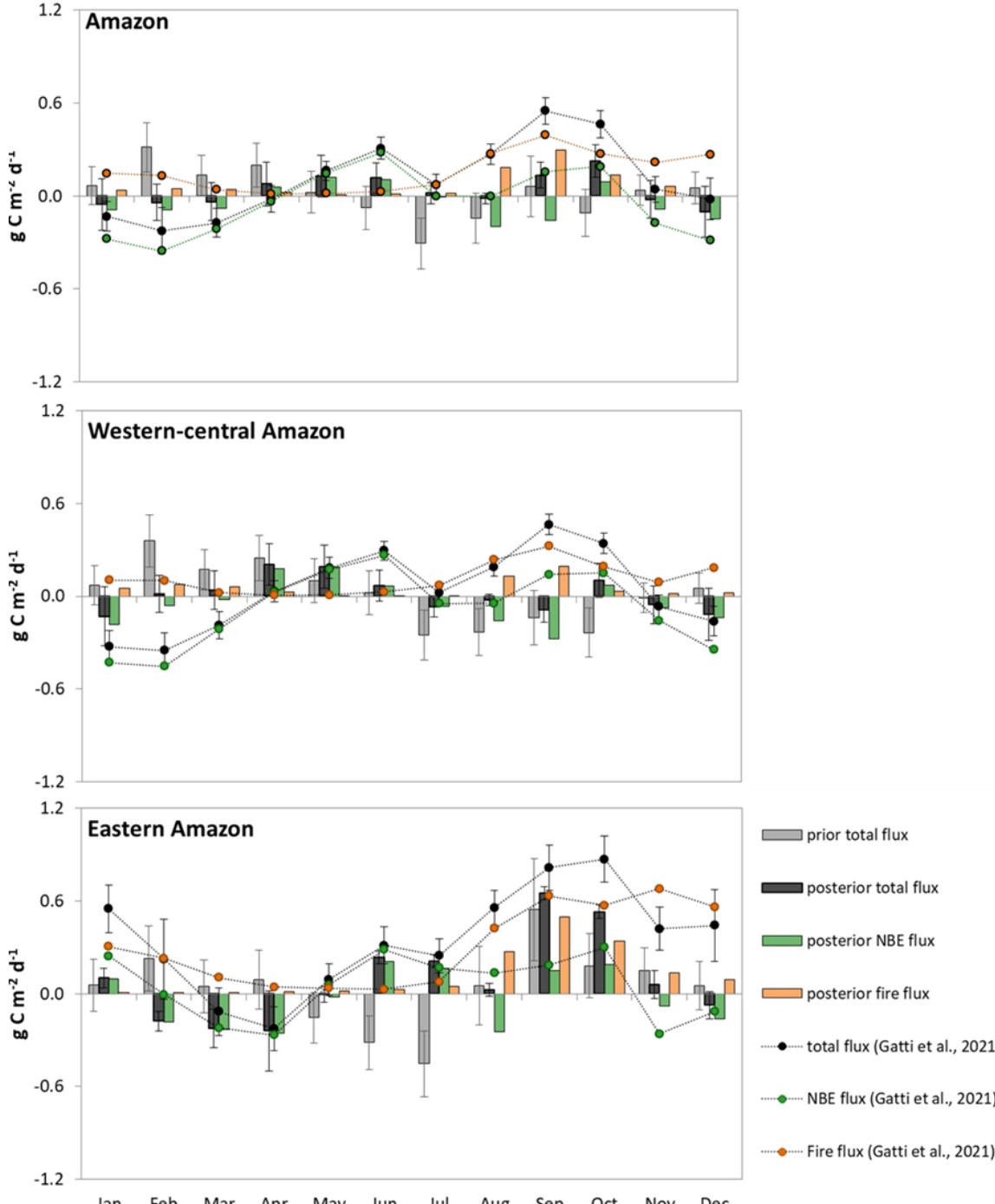

Figure A20. Comparison of monthly mean C fluxes from inverse modelling using Amazon vertical profile observations and C fluxes based the vertical profile observations calculated by mass balance technique from Gatti et al. (2021), for the period between 2010 and 2018.