# Peer review of "Atmospheric CO2 inversion reveals the Amazon as a minor carbon source caused by fire emissions, with forest uptake offsetting about half of these emissions"

_EGUsphere, 2023_

## Referee Comment (RC1)

**Review of Basso et al.,**

This study analyzes total carbon fluxes, NBE and fire fluxes for the Amazon basin over the period 2010-2018, using $CO_2$ vertical profiles and global flask measurements with a global atmospheric transport model. Their results show the Amazon as a small net source of carbon. These emissions mainly come from the eastern Amazon due to fire emissions, half of which are removed from forest uptake. This study is of interest to the global carbon community, not only for monitoring the Amazon basin, but also for recognizing the need for local measurements and the significance of fires in this region. The manuscript is generally well written and explained but it lacks some details.

**General comments**

Ln.21. What changes are you referring to?

Ln.47. What is the contribution of climate variability versus anthropogenic influences (fire and land use land change (LULC)) to the Amazon forest over the past 40 years? You mention that fire and LULC are the main disturbances for the Amazon forest, but what percentage are we talking about?

Figure 1. It would be interesting to have the outline of the Amazon forest on your map. It is difficult to estimate if, for example, the southeast site is on the border of the Amazon or in the middle of the forest. The 5 Amazon sites could be added in your Figure A14. Additionally, names of the sites could be included in the figure.

Ln. 146. Your simulations are run at almost 6x6 degree horizontal resolution. It is not specified wheter it is used globally as well as regionally. What would be the uncertainties/impacts associated with the coarse resolution of your inversion on your results?

Ln. 184. It is not clear if you have taken the average climatology 2003-2013 of CASA-GFED4 for your 2010-2018 period with a scale to adjust this climatology to your study period. More information should be provided wheter or not a scale was used, and the impact on your posterior fluxes for not having it scale. Additionally, why not using GEOS-Carb CASA-GFED v3, which has a temporal coverage from 2003 to December 2017 closer to your study period than what you used?

Ln. 206 and Ln. 492. It is not well explained how the MODIS burned fraction product is used here, regarding CARDAMON. MODIS burned fraction product has been shown to underestimate information about the burned area compared to the VIIRS product. How do you think this underestimation impacts and biases your results?

Ln. 209. What are the uncertainties associated with the forest biomass removal of the Global Forest Watch?

Figure 3. Posterior total flux estimates with Amazon vertical profile show sources of carbon in the western coastal part of the continent from October to March. This carbon source does not seem to be linked to the fires. What would be the cause? This carbon source is also observe in the annual mean with the posterior estimates and not with the prior estimates.

Ln.317. Positive NBE are observed between April and June for the western-central region, probably caused by decomposition process following years of a burning event. We should observe similar seasonality for the eastern Amazon, with positive NBE following years of burning event, or it does not seem to be the case. Could you explain what could be the differences between the eastern and western regions regarding the NBE and fires seasonality?

Ln. 330. No significant relationships between monthly posterior NBE fluxes and climate variables was observed. However, you mentioned earlier that NBE fluxes could be positive several months following a fire event which is correlated to climate variables. Have you looked at the correlation between NBE fluxes and climate variables with a time lag? Additionally, figures A5 and A6 could be performed for fire and non-fire seasons.

Figure A.11. It would be interesting to have a comparison similar to Figure 2 in order to see the differences between prior monthly mean $CO_2$ mole fractions for prior with CASA and with CARDAMON against respective posterior and observations.

Figure A.12. And Ln. 371. You mentioned having similar variability and flux magnitudes between your NBE and fire estimates compared to the estimates based on Naus et al (2022). It is important to notify some non-negligible differences between both estimates for the annual mean. The differences in fires between both estimates can be significant for the NBE estimates. For example, your NBE can show a

sink of carbon, like in 2010, while the NBE from GFAS and CO_opt show sources of carbon. In 2013, your NBE shows a source while NBE CO_opt shows a sink. Additionally, when both NBEs agree to have a carbon sink, the sink will be larger or not depending on the fires estimates. It is therefore important to note and conclude here that NBE estimates are not only constrained by observations but also by fire estimates, as observed in Peiro et al., (2022, https://doi.org/10.5194/acp-22-15817-2022). This seems to be particularly true for tropical regions where fires are a dominant component.

Details are missing on how you considered and separated deforestation, logging operations and ground-fires in your work and inversion. Could you elaborate?
As detailed in Andela et al., 2022 (DOI: 10.1126/sciadv.abd2713), fires in the Amazon basin can be classified into four different types (deforestation fires, propagating or forest fires, agricultural management, grassland fires occurring in pasture lands at the border of the forest.). In ln. 464 and 480, you mention agricultural and deforestation fires but without further details for your results. Knowing that these different types of fires have different characteristics, combustion, and may or may not be captured by satellite active fire detection, more explanation and detail should appear in the discussion of how these would affect your inversions and your results.

**Technical comments**

Ln. 58. Would suggest a comma after continues.

Ln. 75 and 78: these two sentences could be rearranged in one sentence. Additionally, NBE being a source of C to the atmosphere is repeated twice.

Ln 175. Could you develop what you mean by your posterior errors are likely to be "lower limits"?

Ln. 213. "To estimate the contribution of biomass burning emissions in Amazon" on what, can you precise?

Ln. 250. How was the 0.1 m/month evapotranspiration estimated for CWD?

Section 2.2.6. It has not be explained so far why solar radiation is introduced here and for which purpose.

Figure 2. Quality of the figure should be improved. Are the posterior with or without the Amazon observations?

Figure 5. Resolution of the figure should be improved, particularly Figure 5.a.

Ln. 325. Should be Figure A4 and A5.

Ln. 335. "Table 1", parenthesis might be missing.

---

## Author Comment (AC1)

**Referee 1**

Thanks for your very helpful comments and suggestions. Please find below our answers for each general and technical comment.

*Ln.21. What changes are you referring to?*

Authors: We refers to changes in temperature and land cover caused by deforestation and forest degradation, as we describe in lines 19 and 21. We rephrased as follows to be clear (line 22):

*"These temperature and land cover changes are expected to affect the forests and an important diagnostic of their health and sensitivity to climate variation is their carbon balance."*

*Ln.47. What is the contribution of climate variability versus anthropogenic influences (fire and land use land change (LULC)) to the Amazon forest over the past 40 years? You mention that fire and LULC are the main disturbances for the Amazon forest, but what percentage are we talking about?*

Authors: It is not possible to say how much is the percentage of the contribution of climate variability versus anthropogenic influences in this region, but we add more details about how deforestation and forest degradation (drive by fires caused by anthropogenic activity in association to drier conditions and logging) contributed to the Amazon forest loss over the last years). We rephrase the paragraph adding new citation reporting amount of above ground biomass lost, and how much of these changes are related with deforestation and forest degradation (lines 50-57).

*"Human-induced land use and cover change and forest degradation (drive by fires caused by anthropogenic activity in association to drier conditions and logging) are the main direct disturbances in the Amazon forest (Fawcett et al., 2022; Lapola et al., 2023), resulting in above ground biomass losses of 1.3 (±0.4) PgC (between 2012 and 2019; Fawcett et al., 2022). Kruid et al. (2021) reported from 2003 and 2019, that 56% of the carbon loss in this region was attributed to deforestation, with the remainder (44%) due to forest degradation/disturbance (including fire, natural disturbances, drought-induced tree mortality, edge effects, selective logging, and other extractive activities). Over the past 40 years the Amazon forest loss accounts for around 17% of its total area (MapBiomas, 2020), and degradation (between 1995 and 2017) accounts for around 17% of total forest area (Lapola et al., 2023)."*

*Figure 1. It would be interesting to have the outline of the Amazon forest on your map. It is difficult to estimate if, for example, the southeast site is on the border of the Amazon or in the middle of the forest. The 5 Amazon sites could be added in your Figure A14. Additionally, names of the sites could be included in the figure.*

Authors: Thanks for the suggestion. We added the outline of the Amazon region on all our maps. In addition, we added the name and location of the sites to all these figures cited before, including the figure A14 (now Figure A6 in the revised manuscript version).

*Ln. 146. Your simulations are run at almost 6x6 degree horizontal resolution. It is not specified wheter it is used globally as well as regionally. What would be the uncertainties/impacts associated with the coarse resolution of your inversion on your results?*

Authors: We now make clear that we did a global inversion (line 163):

*"The forward and adjoint model simulations were carried out globally at 5.6° x 5.6° horizontal resolution, with 60 vertical levels up to 0.1 hPa."*

Regarding the uncertainties/impacts of the model coarse resolution, it could be possible that in some sites/regions where the $CO_2$ concentrations could have a stronger dependence of

local process/effects are not well represented by the coarse transport model resolution. Also, it is possible that our estimates may could not report small changes on carbon fluxes in local scales due the coarse resolution.

We have not formally investigated how spatial resolution affects uncertainties of flux estimates and it would exceed the remit of this study. We have quantified and documented the uncertainties of the approach, with uncertainties including representation uncertainty, which in our view is key for this study.

Independently with regards to spatial resolution of the transport model underlying the inversions: it is not clear whether a resolution which exceeds the spatial density of the data will reduce uncertainties. The resolution we use is commensurate with the spatial density of the data.

Previous $CH_4$ inversion estimates using TOMCAT model with inversions at 2.8° and 5.6° resolution and with GOSAT data (Wilson et al., 2021), and the authors reported that the results were robust at both resolutions. Also, was previously investigated the effects of resolution in the inversions and the authors found that they are smaller than the observation uncertainty in most cases (Wilson et al., 2014). We included this point in the text (lines 164-166):

*"Although we did not investigate the uncertainties of the coarse resolution in our estimates, previous $CH_4$ inversion estimates using TOMCAT model with inversions at 2.8° and 5.6° resolution and assimilating GOSAT data showed that the results were robust at both resolutions (Wilson et al., 2021)."*

*Ln. 184. It is not clear if you have taken the average climatology 2003-2013 of CASA-GFED4 for your 2010-2018 period with a scale to adjust this climatology to your study period. More information should be provided wheter or not a scale was used, and the impact on your posterior fluxes for not having it scale. Additionally, why not using GEOS-Carb CASA-GFED v3, which has a temporal coverage from 2003 to December 2017 closer to your study period than what you used?*

Authors: There are many possibilities that can be used as priors for land-atmosphere exchange fluxes as well as fire fluxes. We use CASA and an average climatology so that the interannual variation in our estimates are driven by the variation observed in the $CO_2$ observations and not by the prior emissions. We didn't apply any scaling factor to adjust this climatology to our studied period. Whilst applying a scaling factor may have slightly changed the mean value of the global prior biosphere flux, the applied prior uncertainty around our prior estimation (1.1 and 0.6 PgC $y^{-1}$ for land and ocean global flux uncertainties) was larger than any change in the mean value would be. Considering the uncertainty in the biosphere prior flux distribution the absence of any factor to adjust the climatology would result in a very small impact in our results.

Regarding the choice of CASA-GFED version. We have been working with CASA-GFED and based on simulations we know that e.g. seasonal cycles at most NOAA sites are well predicted and thus we opted to use this version. There are various options one could use for these priors and from our experience most important is that uncertainties used are realistic in these inversions. The various 'products' do differ and which one comes closer to reality is difficult to assess.

*Ln. 206 and Ln. 492. It is not well explained how the MODIS burned fraction product is used here, regarding CARDAMON. MODIS burned fraction product has been shown to underestimate information about the burned area compared to the VIIRS product. How do you think this underestimation impacts and biases your results?*

Authors: We added a more detailed explanation of how MODIS burned fraction product is used in CARDAMOM (lines 232 to 235):

*"Emissions are determined assuming as the product of the MODIS burned fraction input, the simulated biomass pools (labile, foliage, roots, wood, litter and soil) and tissue specific combustion completeness (CC) parameters. The CC parameters are estimated on a per-pixel basis as part of the CARDAMOM process. As part DALEC's fire model, a fraction of the burned but not combusted biomass pools undergoes mortality (tissue resilience) resulting in the generation of litter. For details see Exbrayat et al. (2018)."*

As described in the methodology, we use the fire fraction from the prior flux to estimate the CO flux from fire. As discussed in the text, the absence of burned area in the grid cell will result in no fire emissions, overestimating our posterior fire emissions and our total carbon emissions, which could be part of the difference to the fire emissions observed by Gatti et al. (2021) using a mass balance technique. To clarify we rephrase how we use the burned area fraction in methodology and discussion (now lines 606 to 609):

*"Also, the difference could also be related to the burned areas fraction from the prior flux (from GFED V4.1s) which we multiplied to the CO total flux in each grid cell to derive the CO fire emissions in our inversion, in the absence of burned area fraction will result in no fire emissions in the area, consequently underestimating carbon emissions from fire in this region."*

As with all prior estimates what is in our understanding central is that uncertainties are well characterized, whether the fire product is VIIRS or MODIS based or based on yet another 'product'.

*Ln. 209. What are the uncertainties associated with the forest biomass removal of the Global Forest Watch?*

Authors: The CARDAMOM estimated biomass removals associated with Global Forest Watch (GFW) have two basic sources. First, the uncertainty associated with the available biomass for harvest, and second due to errors in the forest cover loss estimate itself. Uncertainty in the former is propagated explicitly using CARDAMOM's ensemble-based estimates in biomass stocks. Uncertainty in the latter is neglected due to insufficient information on uncertainties associated with Global Forest Watch. Deforestation events smaller than the spatial resolution of GFW. As a result, forest losses due to degradation are missed, where biomass is lost without detectable change in canopy cover (Milodowski et al., 2017). However, GFW has been shown to provide robust detection of forest loss events above its resolution threshold. GFW is also relatively unique in providing a long timeseries of temporally consistent estimates.

Reference: Milodowski, D., Mitchard, E., & Williams, M. (2017). Forest loss maps from regional satellite monitoring systematically underestimate deforestation in two rapidly changing parts of the Amazon. *Environmental Research Letters*. https://doi.org/10.1088/1748-9326/aa7e1e

*Figure 3. Posterior total flux estimates with Amazon vertical profile show sources of carbon in the western coastal part of the continent from October to March. This carbon source does not seem to be linked to the fires. What would be the cause? This carbon source is also observe in the annual mean with the posterior estimates and not with the prior estimates.*

Authors: It is a good question. Unfortunately, inversions do not attribute causes to the fluxes it estimates. As the cause is unlikely fires it must be related to vegetation and soils. A

reason for carbon loss could be the effect of climate warming and possibly changes in precipitation on vegetation and soils (e.g. publications by Andrew Nottingham).

*Ln.317. Positive NBE are observed between April and June for the western-central region, probably caused by decomposition process following years of a burning event. We should observe similar seasonality for the eastern Amazon, with positive NBE following years of burning event, or it does not seem to be the case. Could you explain what could be the differences between the eastern and western regions regarding the NBE and fires seasonality?*

Authors: Western-central and eastern Amazon regions have different seasonality and magnitude in both carbon fluxes (from fires and NBE). As mentioned by the reviewer the positive NBE between April and June could be related with decomposition emissions in the western-central region. NBE is also positive during June and July in the eastern region (although the magnitude of emissions is lower than in the western-region), which could also be related with decomposition process. It is important to highlight that the climatic variables seasonality is also different in these regions, with wet, dry and transition periods happen not simultaneously in both regions, affecting the seasonality of fire and NBE fluxes. One possible reason for the difference in the positive NBE is probably related with decomposition process between both regions, as the eastern region has lower forest cover. The eastern region is the area with higher levels of accumulated deforested and land cover change areas. Also, the eastern region includes the Amazon biome to Cerrado (savannas) transition region, where higher fire emissions occurs and less carbon uptake is taken up, as can see now in the updated map figures including the Amazon region (based on Eva and Huber, 2005). According to these authors, the southern border of the Amazon region is characterized by a steady transition from an essentially forested landscape (the Amazon lowland rainforest) to a mainly non-forested landscape, where predominates open vegetation types, such as savannas (campos cerrados and alluvial flooded savannas), savanna woodlands (cerradão) and other mainly scrubby vegetation types. We added in the text that we also observe positive NBE fluxes in the eastern Amazon, and this more detailed description of the southeastern Amazon region. (lines 359-365):

*"In the eastern region we also estimate positive NBE fluxes (during June and July) that could be related to the decomposition process, but these emissions have lower magnitude than the observed in the western-central region. We highlight that the southern border of the Amazon region is characterized by a steady transition from the Amazon lowland rainforest to a mainly non-forested landscape, where predominates open vegetation types, such as savannas (campos cerrados and alluvial flooded savannas), savanna woodlands (cerradão) and other mainly scrubby vegetation types (Eva and Huber, 2005) Also, includes deforested areas with land use change conversion (Figure A6)."*

*Ln. 330. No significant relationships between monthly posterior NBE fluxes and climate variables was observed. However, you mentioned earlier that NBE fluxes could be positive several months following a fire event which is correlated to climate variables. Have you looked at the correlation between NBE fluxes and climate variables with a time lag? Additionally, figures A5 and A6 could be performed for fire and non-fire seasons.*

Authors: Yes, we looked at the correlation between NBE fluxes and the climate variables with a time lag (using 1, 2 and 3 months of lag), but we didn't find significant correlation and improvement in the results. So, we decided to show in the paper the results without any time lag. We agree that it would be good to add the information in the text that we also investigate the correlations for NBE with a time lag (lines 381-383):

*"We also investigate the correlation between NBE fluxes and climate variables with a time lag (one, two and three months of lag) but no significant correlation was observed."*

We also looked for correlations using wet and dry periods, but didn't find significant correlations for NBE and the results for fire fluxes during dry season were similar to using the whole year. It is important to highlight that climate conditions in Amazon were not homogeneous, varying regionally. The wet and dry season happens during different months, so doing the analysis for two main regions like we did could not represent all this variability.

*Figure A.11. It would be interesting to have a comparison similar to Figure 2 in order to see the differences between prior monthly mean CO2 mole fractions for prior with CASA and with CARDAMON against respective posterior and observations.*

Authors: Thanks for the suggestion, we have included a plot of this in the appendix (figure A15) and the mean difference between model and observed mole fractions in table A2. We also add a briefly discussion on lines 419 to 421:

*"We also observed a large improvement after the assimilation of observations in the model as for the inversions using CASA as prior flux estimates (Figure A15 and Table A2)."*

*Figure A.12. And Ln. 371. You mentioned having similar variability and flux magnitudes between your NBE and fire estimates compared to the estimates based on Naus et al (2022). It is important to notify some non-negligible differences between both estimates for the annual mean. The differences in fires between both estimates can be significant for the NBE estimates. For example, your NBE can show a sink of carbon, like in 2010, while the NBE from GFAS and CO_opt show sources of carbon. In 2013, your NBE shows a source while NBE CO_opt shows a sink. Additionally, when both NBEs agree to have a carbon sink, the sink will be larger or not depending on the fires estimates. It is therefore important to note and conclude here that NBE estimates are not only constrained by observations but also by fire estimates, as observed in Peiro et al., (2022, https://doi.org/10.5194/acp-22-15817-2022). This seems to be particularly true for tropical regions where fires are a dominant component.*

Authors: We agree that NBE estimates could also have some influence from the fire emissions. As we show here the posterior fluxes are much more similar to the prior fluxes in Amazon region in absence of measurements over this region assimilated in the inversions. So, it is expected that the $CO_2$ flux estimates from Peiro et al. (2022) have more dependence on the prior emissions in the Amazon region than ours. In general, the number of in situ observations is particularly low in the tropics, which could estimate posterior fluxes with more dependence of the prior fluxes in this region. But in our inversion we reduced this effect by assimilating the Amazon vertical profile observations.

Also, it is important to highlight some points regarding our sensitivity test. First, we didn't use the $CO_2$ fire emissions from the CO flux optimized as prior in our $CO_2$ inversions, as done in the paper mentioned by the reviewer. We just subtracted the fire $CO_2$ flux (based on the CO independent inversions) from our total $CO_2$ posteriori flux. Which means that if both $CO_2$ fire emissions (based on the CO independent inversions) have larger differences, these will be reflected in the NBE flux as can be seen in the figure A16 b and c.

Regarding the difference in the carbon fire emissions (and NBE) it is important to highlight that both CO inversions have differences in the atmospheric transport model but also in the conversion of CO from fire to $CO_2$ fire emissions. As described in the methodology in the CO inversion using TOMCAT/INVICAT we used the emission factor to convert the CO emissions to $CO_2$ using the emission ratio from the Amazon vertical profiles reported in Gatti et al. (2021). While in the conversion to $CO_2$ from the CO fire emissions from Naus et al.

(2022) was used the CO:CO$_2$ ratios based on GFAS emission factors for each grid cell. Thus, differences in estimates from both approaches were expected. It is possible that if these CO$_2$ fire emissions were used as prior in the CO$_2$ inversion, rather than subtracted from the total CO$_2$ flux optimized these differences could be reduced since the CO$_2$ observations will adjust the posteriori fluxes, but future tests should be done. We add in lines 437-442 that the variations in the results could also be related to the CO:CO$_2$ emission factor used, and that future analysis could be done using both CO$_2$ fire emissions (based on the two different CO optimized fluxes) as prior fluxes to investigate the dependence of fire emissions in the NBE optimization:

*"Both CO inversions assimilated the same MOPITT observations, but have variations in inversion methodology, model transport and in the emission factor to convert CO flux from fires to CO$_2$ flux. Some difference in both estimates (for both fire and NBE fluxes) could be related with these differences in both approaches. To get a true independent estimate of NBE from another model, it would need to produce posterior estimates of both total carbon and fire carbon. Also, both CO$_2$ fire emissions (based on the two different CO optimized fluxes) could be used as prior in a future CO$_2$ inversion to investigate the dependence of the fires estimates in the NBE optimization."*

*Details are missing on how you considered and separated deforestation, logging operations and groundfires in your work and inversion. Could you elaborate? As detailed in Andela et al., 2022 (DOI: 10.1126/sciadv.abd27 13) , fires in the Amazon basin can be classified into four different types (deforestation fires, propagating or forest fires, agricultural management, grassland fires occurring in pasture lands at the border of the forest.). In ln. 464 and 480, you mention agricultural and deforestation fires but without further details for your results. Knowing that these different types of fires have different characteristics, combustion, and may or may not be captured by satellite active fire detection, more explanation and detail should appear in the discussion of how these would affect your inversions and your results.*

Authors: In our CO inversion we are quantifying the total CO flux, thus what type of fires are there will not make a difference. However, the kind of fire could be relevant when we convert the CO flux to CO$_2$ flux using the scaling factor, since this factor is different for different types of fire. In our conversion of CO flux to CO$_2$ flux, this information is not necessary, because as described in the methodology the emission factor used (CO$_2$:CO ratio) was based on the Amazon vertical profiles ratio reported by Gatti et al. (2021). This ratio is based on the atmospheric concentration, which means that is a result of all the emissions from all the different fire categories already mixed in the atmosphere.

**Technical comments**
*Ln. 58. Would suggest a comma after continues.*
Authors: Done

*Ln. 75 and 78: these two sentences could be rearranged in one sentence. Additionally, NBE being a source of C to the atmosphere is repeated twice.*
*Authors: We rephrase as suggested.*

*Ln 175. Could you develop what you mean by your posterior errors are likely to be "lower limits"?*
Authors: We rewrote this (line 197):

*"Considering that this iterative method estimates the inverse of the Hessian (the second derivative) of the cost function and the off-diagonal elements of the posterior covariance matrix are not included, the posterior errors included here are estimates, with their own remining uncertainties (Bousserez et al., 2015)."*

*Ln. 213. "To estimate the contribution of biomass burning emissions in Amazon" on what, can you precise?*

        Authors: Done

*Ln. 250. How was the 0.1 m/month evapotranspiration estimated for CWD?*

        Authors: The value 0.1 m/month used for evapotranspiration is an approximated value (not estimated in this study), based on the evapotranspiration value reported by Aragao et al. (2007, GRL). Although this is an approximated value the diagnostic has been used in other previous published analysis (Tavares et al., 2023, https://doi.org/10.1038/s41586-023-05971-3; Phillips et al, 2009, https://doi.org/10.1126/science.1164033, as example).

*Section 2.2.6. It has not be explained so far why solar radiation is introduced here and for which purpose.*

        Authors: We add an explanation of which is the purpose in lines 114 to 116:

        *"In Section 2 we describe the inverse modelling approach and describe the observations used, in Sections 3 and 4 we discuss our results, analyze the drivers of $CO_2$ fluxes (as cumulative water deficit, temperature, solar radiation and burned area), cross-validate our model mole fractions with independent Amazon observations and compare our estimates with other previous published Amazonian estimates, mainly with estimates using an air column mass balance technique."*

*Figure 2. Quality of the figure should be improved. Are the posterior with or without the Amazon observations?*

        Authors: The posterior are with the Amazon observations; we add this information in the figure legend. Also, we improved the quality of the figure.

*Figure 5. Resolution of the figure should be improved, particularly Figure 5.a.*

        Authors: Done

*Ln. 325. Should be Figure A4 and A5.*

        Authors: Done

*Ln. 335. "Table 1", parenthesis might be missing.*

        Authors: Done

---

## Author Comment (AC2)

Thanks for your very helpful comments and suggestions. Please find below our answers for each general and technical comment. Just to clarify, we didn't use CO observations from Amazon vertical profiles in our inversions. For CO inversions using TOMCAT model we used MOPPIT data.

**General Comments.**

*1. The model setup needs to be described in more detail to help readers visualize the experiment setup. The spin-up used for simulations, and any other relevant details should be provided. Additionally, since vertical profiles are being assimilated, discussing and providing validation plots regarding vertical transport would be beneficial. It is important to consider how much uncertainty in vertical transport might affect the results. The uncertainties and impacts associated with the coarse resolution of the inversion on the results should also be discussed. Additionally, it would be useful to know if all sites used in the inversion have full data coverage or if some have discontinuous data. The information on the data period should also be included in Table A1.*

Authors: As with all $CO_2$ inverse model studies, our ability to assess and quantify flux uncertainties associated with transport errors is very limited and still represents a major deficit in our field. Nonetheless, we appreciate the reviewer's concern and point out the following:

- A previous study with simulations of sulfur hexafluoride ($SF_6$) and other species comparing different transport models investigated some of the large-scale transport characteristics (Patra et al., 2011), and shows that TOMCAT in general performed well slightly overestimating the $SF_6$ inter-hemispheric gradient compared to observations, but within the bounds of other transport models. We included this point in the text (lines 159-162):

*"A previous study with simulations of sulfur hexafluoride ($SF_6$) and other species comparing different transport models investigated some of the large-scale transport characteristics (Patra et al., 2011), and shows that TOMCAT in general performed well, slightly overestimating the $SF_6$ inter-hemispheric gradient compared to observations, but within the bounds of other transport models."*

- We've performed cross-validation analyses using tall tower (data from ATTO tower) and aircraft data from the Amazon Basin (vertical profiles near Manaus, MAN), not used as observational constraints on $CO_2$ flux. The figure A 17 and A18 shows the monthly mean bias between model (prior or posterior) and the ATTO observations. Measurements from ATTO are hourly and here we used the daily mean based on the measurements between 13h-17h UTC. Although the ATTO tower measurements were for 80m height, in general we found a good agreement between model and observations, with a reduction of the bias after the inversion from 0.9 (range of -3.9 to 7.7) ppm to 0.3 (range of -5.3 to 4.7) ppm (t-test: p <0.05). We highlight that the ATTO timeseries started in 2012, but the measurements in the first two years have some gaps to constrain the monthly means. Also, for the year 2015 we remove from the comparison the months without vertical profile data assimilated in the inversion. In addition, we compare the model concentrations to the aircraft vertical profiles in MAN above 3.5km and below 1.5km, as showed in the figure below. The data record for the same period of our inversions is short (MAN vertical profiles data are available for 2017 and 2018), but in general we found a reduction in the bias between model and observations after the inversions (for the mean below 1.5km height: from -0.3 [-6.5 to 6.6] ppm to 0.2 [-4.3 to 5.0] ppm and t-test: p = 0.17; and for the mean above 3.5km height: from -0.1 [-3.1 to 2.1] ppm to -0.4 [-1.9 to 0.5] ppm and t-test: p = 0.13). We also found a reduction on the mean bias of the difference between the mean below 1.5km and the vertical profile free troposphere (from -0.2 [-5.4 to 7.6] ppm to 0.6 [-4.2 to 5.6] ppm and t-test: p = 0.08). We added in the text a new topic to discuss the

comparison with the independent measurements (subsection 3.4) and the figures below in the Appendix (figures A17 and A18).

- Regarding the uncertainties due to the coarse resolution, we have not formally investigated how spatial resolution affects uncertainties of flux estimates and it would exceed the remit of this study. Independently with regards to spatial resolution of the transport model underlying the inversions: it is not clear whether a resolution which exceeds the spatial density of the data will reduce uncertainties. Previous $CH_4$ inversion estimates using TOMCAT model with inversions at 2.8° and 5.6° resolution and with GOSAT data (Wilson et al., 2021), and the authors reported that the results were robust at both resolutions. Also, we previously investigated the effects of resolution in the inversions and the authors found that they are smaller than the observation uncertainty in most cases (Wilson et al., 2014). We included this point in the text (lines 164-166):

*"Although we did not investigate the uncertainties of the coarse resolution in our estimates, previous $CH_4$ inversion estimates using TOMCAT model with inversions at 2.8° and 5.6° resolution and assimilating GOSAT data showed that the results were robust at both resolutions (Wilson et al., 2021)."*

We have quantified and documented the uncertainties of the approach, with uncertainties including representation uncertainty, which in our view is key for this study.

*2. It is not clear whether the inversion is performed globally or regionally. If it is performed globally, it would be useful to compare the simulated growth rate with the observed global growth rate as a metric to test the modeled growth. (e.g., Fig. 5d in Chandra et al., 2022: https://acp.copernicus.org/articles/22/9215/2022/).*

Authors: We add in line 164 the information that the inversion was done globally:

*"The forward and adjoint model simulations were carried out globally at 5.6° x 5.6° horizontal resolution, with 60 vertical levels up to 0.1 hPa."*

We also include a comparison between our global monthly mean mole fraction based on posterior flux and the NOAA monthly mean mole fraction from marine surface sites (Figure A1 and lines 313-315).

*"Estimated posteriori $CO_2$ mole fractions have a similar magnitude and positive trend as seen in the observed, also for the global posterior mean mole fraction which follows the global increase in $CO_2$ global mean observed mole fraction (Figure A1)."*

*3. The author used the same profiles used in the inversion for the evaluation of atmospheric inversion. However, evaluating the inversion using independent observations not used in the inversion, e.g., HIPPO, AToM, CONTRAIL, or other regular aircraft measurements from other campaigns, would be more rigorous.*

Authors: As discussed in the previous comment (please see above discussion), we have performed cross-validation analyses using tall tower (data from ATTO tower) and aircraft data from vertical profiles in Amazon region (MAN), not used as observational constraints on $CO_2$ flux. We added in the text a new topic to discuss the comparison with the independent measurements (subsection 3.4) and the figures A17 and A18 in the Appendix.

*4. It is unclear which fire emissions are used in the calculation. For example, in line#165, it is mentioned that the fire emissions are optimized in the CO2 inversion estimate. Then, Section 2.2.3 discusses the optimization of carbon fire emissions from INVICAT using MOPITT CO. It*

*is unclear whether these two are independent or the same. Additionally, it is unclear which fire emissions are used in the calculations of the Amazonia carbon budget. A comparison plot of prior and optimized BB will be helpful to visualize the correction in BB emissions.*

Authors: In our $CO_2$ inversions we estimate the total $CO_2$ fluxes, we didn't split the $CO_2$ emissions in NBE and fire using the $CO_2$ prior fluxes. For estimate the carbon emissions from fires for the Amazon carbon budget we use carbon fire emissions from the optimized CO total flux from INVICAT using MOPITT CO as described in the methodology. These inversions ($CO_2$ and CO using TOMCAT/INVICAT) are independent. Also, in the CO inversion was not assimilated any vertical profile data in the Amazon region. We added this information in the text to make it clear (lines 111-113, 240-243).

*"We also estimate carbon emissions from fires to constrain the Amazon carbon budget using flux estimates from an independent global inverse modeling based on atmospheric carbon monoxide (CO) measured from space, and relate the carbon fluxes (total, fire and NBE) to climate controls."*

*"To estimate the contribution of biomass burning emissions in Amazon total carbon emissions, we estimated carbon fire emissions with an independent inversion with TOMCAT/INVICAT by assimilating total column carbon monoxide (CO) values from MOPITT radiometer data (V8) on the TERRA satellite (Deeter et al., 2019) globally. Note, that in this inversion was not assimilated any vertical profile data for the Amazon region."*

**Technical Comments**

*1. At line#231, the biomass burning emission ratios are given as 16 ppm CO/ppm CO2. IS the unit correct (or ppb/ppm)? And this number is not found in the cited reference.*

Authors: Thanks for point this. The units of both CO and $CO_2$ are correct in ppm, but should be 16 ppmCO2/ppmCO. We correct that in the text. We decided to use both (CO and $CO_2$) in the same unit to make easy for the reader. The reference of this number is also correct, since we calculated a mean of the emission ratios reported by Gatti et al. (2021) for each one of the four sites (ALF CO:CO2 = 53.4 ± 9.9 (1σ variability); SAN CO:CO2 = 55.5 ± 14.7; RBA CO:CO2 = 73.2 ± 15.1; and TAB_TEF CO:CO2 = 71.6 ± 17.2, which gives a mean of CO:CO2 = 63.425, which translates into (1000/63.425) = ~16 ppmCO2/ppmCO.

*2. The country boundaries in Figure 1 make it too messy. It would be better to remove the country boundaries and include the mask of the Amazon. Including the abbreviation of site names would also be helpful.*

Authors: We edited the figure as the reviewer suggested. We add a blue contour for the Amazon area based on Eva and Huber (2005) and the Amazon sites labels. We didn't add the labels for all NOAA stations because will be a lot of information and make harder to read. We also corrected the position (latitude and longitude) of few stations that was wrong in the submitted version of the paper.